# AUTOMATIC FORECASTING VIA META-LEARNING

## ABSTRACT

In this work, we develop techniques for fast automatic selection of the best forecasting model for a new unseen time-series dataset, without having to first train (or evaluate) all the models on the new time-series data to select the best one. In particular, we develop a forecasting meta-learning approach called AUTOFORECAST that allows for the quick inference of the best time-series forecasting model for an unseen dataset. Our approach learns both forecasting models performances over time horizon of same dataset and task similarity across different datasets. The experiments demonstrate the effectiveness of the approach over state-of-the-art (SOTA) single and ensemble methods and several SOTA meta-learners (adapted to our problem) in terms of selecting better forecasting models (i.e., 2X gain) for unseen tasks for univariate and multivariate testbeds.

## 1 INTRODUCTION

Accurate time-series forecasting at scale is critical for several industrial domains such as cloud computing Poghosyan et al. (2021), supply chain Abbasimehr et al. (2020), energy Abdallah et al. (2021), and finance Oreshkin et al. (2020). Most of the current time-series forecasting solutions are built by experts and require significant manual effort in model construction, feature engineering, and hyper-parameter tuning Bergstra et al. (2011). Hence, they do not scale while generating high-quality forecasts for a wide variety of applications. Moreover, there is no learning scheme that is uniformly better than all other learning schemes for all problem instances Wolpert (1996) (see Appendix A). A naïve approach would be, given a new dataset, we search over thousands of models to select the best forecasting model for the problem at hand. However, this approach is practically infeasible due to the untenable time burden for every new problem.

In this work, we formulate the problem of automatic and fast selection of the best time-series forecasting model as a meta-learning problem. Our solution avoids the infeasible burden of first training each of the models and then evaluating each one to select the best model for a new unseen time-series dataset, or even a new time window within a non-stationary dataset. A practically important desideratum for any solution to this problem is that once the meta-learner $\mathcal{L}$ is trained in an offline manner using a large corpus of time-series data, then we can use it to *quickly* infer the best forecasting model. The quick inference requirement of this new problem, makes it challenging to solve, yet practically important. Our meta-learner $\mathcal{L}$ is trained on the models' performances on historical datasets and the time-series meta-features of these datasets.

We emphasize that the new *time-series forecasting model selection meta-learning problem* introduced in this paper has several unique characteristics and challenges compared to previous related meta-learning problems, e.g., Finn et al. (2017); Rusu et al. (2019); Wistuba et al. (2018). First, existing time-series forecasting models have different designs and different assumptions around the characteristics of time-series (e.g., probabilistic, seasonal, traditional, etc.). Therefore, different models perform differently depending on the characteristics that each dataset exhibits. Thus, capturing the similarity among different datasets needs careful selection of representative time-series meta-features. Second, the new meta-learning approach should capture the temporal variations of the models' performances over different time windows of the dataset, i.e., the best time-series forecasting model for time window $w_t$ is not necessarily the best model for a subsequent time window $w_{t+k}$ (see Figures 5-6 in Appendix B). Third, the number of available time-series forecasting models is large and thus training each forecasting model and then evaluating the suitability of each in inference leads to an unacceptable time burden for most real-world scenarios. These challenges motivate the need for our proposed approach.

To solve the problem of *automatic* time-series forecasting model selection, we propose a temporal meta-learning approach, called AUTOFORECAST that makes time-series forecasting model selection. The intuition of our approach is learning both models' performance evolution over time horizon for same dataset (via our "temporal meta-learner") and task similarity across different datasets (via our "general meta-learner" and proposing novel time-series meta-features that capture their characteristics). In particular, we train our meta-learner using a large model space which has over 320 forecasting models (Section 5.1). We also generate more than 800 meta-features that represent five different types of meta-features which are (simple, statistical, information theoretic, spectral-based, and landmarker), which reflect various characteristics of the time-series dataset (Section 3). We also consider diverse datasets so our meta-learning model can be generalized on new tasks (Table 1) and release the corpus of datasets, along with their meta-features and the performances across hundreds of models. Given a new (unseen) dataset, AUTOFORECAST automatically determines the best forecasting model among a large space of models, without the need to train and evaluate any of the different forecasting models on this new dataset.

The experiments demonstrate the effectiveness of our proposed approach where we validate our meta-learning approach on both univariate and multivariate testbeds. In particular, we show the superiority of our approach over the state-of-the-art (SOTA) time series forecasting models Salinas et al. (2020); Wang et al. (2019); Taylor & Letham (2018); Montero-Manso et al. (2020); Liaw et al. (2002) and different meta-learning approaches Kadioglu et al. (2010); Nikolić et al. (2013); Zhao et al. (2020). Across all datasets, AUTOFORECAST achieves 2X gain in selecting the best forecasting model. Moreover, AUTOFORECAST yields a significant reduction in inference time over the naïve approach, i.e., across all datasets, AUTOFORECAST has a 42X median inference time reduction. We find empirically that no single forecasting model triumphs in more than 0.7% of the datasets, thus motivating the need for automatic model selection for new time-series datasets.

**Summary of Main Contributions.** The key contributions of this work are as follows:

1. **Problem Formulation**: We propose a novel meta-learning approach that predicts the best model for new (unseen) forecasting tasks where each model is composed of time-series specific parameters along with traditional meta-learning model components.

2. **Temporal Learning of Performances**: We propose a time-series meta-learner that learns the models' performances evolution over time windows of the datasets. Our meta-learner has two main sub-learners — the time-series (temporal) meta-learner and the general meta-learner that are designed for different data types with different time dependencies.

3. **Specialized Meta-features for Time-series Forecasting**: We design novel time-series landmarker meta-features to capture the unique characteristics of a time-series dataset toward effectively quantifying task similarity.

4. **Efficiency and Effectiveness**: Given a new time-series dataset, AUTOFORECAST selects the best performing forecasting algorithm and its associated hyperparameters without requiring any model evaluations, while incurring negligible run-time overhead. Through extensive experiments on our benchmark testbeds, we show that selecting a model by AUTOFORECAST outperforms SOTA meta-learners and popular forecasting models.

5. **Benchmark Data:** We release our meta-learning database corpus (348 datasets), performances of the 322 forecasting models, and meta-features for the community to access it for forecasting model selection and to build on it with new datasets and models.[1]

## 2 RELATED WORK

**Meta-learning in Time-series Forecasting:** There are few works that considered meta-learning for time-series analysis Hooshmand & Sharma (2019); Ribeiro et al. (2018); Pan et al. (2020). The works Hooshmand & Sharma (2019); Ribeiro et al. (2018) applied a neural network time-series forecasting model trained on a source (energy) dataset and fine-tuned it on the target (energy) dataset. However, these works did not consider using meta-learning for the general problem of forecasting model selection that we consider in our current work. There also exist few works that have explored model selection problem using ensemble learning Ye & Dai (2018); Talagala

---

[1]Anonymous URL for our database:
https://drive.google.com/drive/folders/1K1w1Ida5Cr15b5Fhidax-i-fNpWZjvet

et al. (2018); Vaiciukynas et al. (2020). However, their problem domain of ensemble learning is different from our problem of model selection. That is due to the fact that ensemble learning constitutes building multiple models for the same task and does not in itself involve learning from prior experience on other tasks. In contrast to those works, AUTOFORECAST can select among any (heterogeneous) set of methods. Finally, there is a line of work that considered empirical analysis for performance estimation Bergmeir & Benítez (2012); Cerqueira et al. (2020) and model selection Cerqueira et al. (2021) in time-series forecasting. However, these works have several distinctions from our work which are the need of evaluating all forecasting models in inference and providing an analysis of the ranking ability of performance estimators without having a meta-learner that learns how to automatically select the best model.

**Few-shot Learning & Transfer Learning:** Few-shot learning has been recently leveraged for automating machine learning pipeline Ravi & Larochelle (2016); Snell et al. (2017); Oreshkin et al. (2020). In particular, the works Ravi & Larochelle (2016); Snell et al. (2017) investigated different problems outside the domain of time-series forecasting. The work Oreshkin et al. (2020) applied meta-learning for zero-shot univariate time series forecasting. However, that work has the limitations of focusing on solving the cold start problem (i.e., learning model parameter initialization that generalizes better to similar tasks) which is different from our forecasting model selection problem, considering different models from the same N-BEATS architecture Oreshkin et al. (2019), and tackling only univariate time-series datasets. We emphasize that our framework can use N-BEATS as one forecasting algorithm in our model space. Finally, there exist few works that applied transfer learning for time series classification (TSC) Fawaz et al. (2018); Abdallah et al. (2021); Wen & Keyes (2019); Narwariya et al. (2020). These works however have two distinctions from our work. First, they transfer the learned network's weights to another network that is also trained on a target dataset. Second, the TSC problem is different from our forecasting problem that we consider here.

## 3 PROBLEM FORMULATION

**Meta-learning Components:** We address the problem of model selection for time-series forecasting via the meta-learning approach. Our proposed meta-learner AUTOFORECAST depends on:

- A collection of historical time-series forecasting datasets $\mathcal{D}_{train} = \{\boldsymbol{D}_1, \boldsymbol{D}_2, \cdots, \boldsymbol{D}_n\}$, namely, a meta-train database, where $n$ is the number of the historical datasets in $\mathcal{D}_{train}$. Note that $\boldsymbol{D}_i \in \mathbb{R}^{n_i \times v_i}$, where $n_i$ is the number of observations of the dataset $\boldsymbol{D}_i$ and $v_i$ is the number of variables in $\boldsymbol{D}_i$.
- The forecasting models that define the model space, denoted as $\mathcal{M} = \{M_1, M_2, \cdots, M_m\}$, where $m$ is the size of the model space $\mathcal{M}$.
- For each dataset $\boldsymbol{D}_i \in \mathcal{D}_{train}$, we sample $T$ windows from $\boldsymbol{D}_i$, where each sample window $w_t$ from dataset $\boldsymbol{D}_i$ has length $|w_t|$ ( smaller than the dataset length); See Appenidx C.1.

**Model Design**: For our forecasting model selection problem, we define our model as follows.

**Definition 1.** *A model $M_i \in \mathcal{M}$ is given by the tuple $M_i = (a_i, \mathbf{h}_i, g_i(\cdot))$, where $a_i$ is the forecasting algorithm, $\mathbf{h}_i$ is the hyperparameter vector for the forecasting algorithm $a_i$, and $g_i(\cdot) : \mathbb{R}^{n_i \times v_i} \rightarrow \mathbb{R}^{n_i \times v_i}$ is the time-series data representation.*

We emphasize that $\mathbf{h}_i$ consists of hyper-parameters of the forecasting algorithm (e.g., number of RNN layers in DeepAR Salinas et al. (2020)) and that $g_i(\cdot)$ represents time-series different representations (e.g., the smoothing transformation Kalekar et al. (2004); see Table 9 in Appendix F).

Using the defined meta-train database, the model space, and the sampling windows, we now introduce the performance tensor.

**Definition 2.** *Given a meta-train database $\mathcal{D}_{train}$ and a model space $\mathcal{M}$, we define the performance tensor $\mathsf{P} \in \mathbb{R}^{T \times n \times m}$ as*

$$\mathsf{P} = \{\boldsymbol{P}_1, \boldsymbol{P}_2, \cdots, \boldsymbol{P}_T\},$$

*where $\boldsymbol{P}_k = (p_k^{i,j}) \in \mathbb{R}^{n \times m}$ and the element $p_k^{i,j} = M_j(w_k(\boldsymbol{D}_i))$ denotes the $j^{th}$ model $M_j$'s performance on the time window $w_k$ of the $i^{th}$ meta-train dataset $\boldsymbol{D}_i$.[2] We thus denote $\boldsymbol{p}_k^i = \begin{bmatrix} p_k^{i,1} & \cdots & p_k^{i,m} \end{bmatrix}$ as the performance vector of all models in $\mathcal{M}$ on time window $w_k$ of $\boldsymbol{D}_i$.*

---

[2]For each time window $w_t$, we have the pair $(\mathbf{x}_{t-|w_t|:t-1}, \mathbf{y}_{t:t+H})$, where the different models are trained on $\mathbf{x}_{t-|w_t|:t-1}$ and evaluated on $\mathbf{y}_{t:t+H}$, where $H$ is the forecasting horizon.

The performance tensor represents the prior experience that the meta-learner will build upon to perform efficiently on the new unseen task (time-series). This motivates us to define our problem.

**Definition 3.** ***Time-series forecasting model selection problem***. *Given a new input task (dataset)* $\boldsymbol{D}_{test}$ *(i.e., unseen time-series forecasting task), the time-series forecasting model selection problem is then stated as follows: for each time window* $w_t$ *in* $\boldsymbol{D}_{test}$*, select the best model* $\hat{M}_t \in \mathcal{M}$ *to employ on that window of* $\boldsymbol{D}_{test}$*. Formally, such selection problem is given by*

$$\hat{M}_t \in \arg\max_{M_j \in \mathcal{M}} M_j(w_t(\boldsymbol{D}_{test})) \, \forall t \in \{1, 2, \ldots, T\}. \tag{1}$$

**Time-series Meta-Features:** A key component of AUTOFORECAST is the extraction of meta-features that aims to capture the important characteristics of a time-series dataset. To achieve such a goal, we extract meta-features for each time-series dataset which we define formally next.

**Definition 4.** Given a time-series dataset $\boldsymbol{D}_i$, we define the meta-features tensor $\mathsf{F}_i = \{\boldsymbol{F}_1^i, \cdots, \boldsymbol{F}_T^i\} \in \mathbb{R}^{T \times d \times v_i}$, where the meta-features matrix $\boldsymbol{F}_k^i \in \mathbb{R}^{d \times v_i}$ denotes the set of the meta features for the time window $w_k$ of the dataset $\boldsymbol{D}_i$, given by

$$\boldsymbol{F}_k^i \triangleq \{\psi(w_k(\boldsymbol{D}_i)) : \psi : \mathbb{R}^{|w_i| \times v_i} \to \mathbb{R}^{d \times v_i}\}, \tag{2}$$

where $\psi(\cdot) : \mathbb{R}^{|w_i| \times v_i} \to \mathbb{R}^{d \times v_i}$ defines feature extraction module in AUTOFORECAST and $d$ denotes the number of the meta-features.[3] We summarize our notations in Table 7 (Appendix C).

**Meta-Features Categories:** The set of meta features in our work that capture the main characteristics of a dataset can be organized into five categories Vanschoren (2018): simple (general task properties), statistical (properties of the underlying dataset distributions), information-theoretic (entropy measures), spectral (frequency domain properties), and landmarker (forecasting models' attributes on the task) features. The idea of the proposed landmarker features is to apply a few of the fast, easy-to-construct time-series forecasting models on a dataset and extract features from (i) the structure of the estimated forecasting model, and (ii) its output performance scores. The complete meta-features list in AUTOFORECAST are explained in Appendix E (Table 8).

## 4 AUTOFORECAST SOLUTION

AUTOFORECAST consists of two-phases: offline training of the meta-learner and online inference that aims at selecting the appropriate model at test time. We argue that running time of the offline training phase is not critical since it is done only once. On the contrary, forecasting model selection for a new time-series dataset should incur small run-time overhead since it is critical for quick selection of the forecasting model. We now explain our meta-learning approach and its components.

### 4.1 META-LEARNING OBJECTIVE AND (OFFLINE) TRAINING

We show the overview of the major components of AUTOFORECAST in Figure 1. We highlight the components transferred from offline to online stage (model selection) in blue; namely, meta-feature extractors $\psi$, feature embedding, time-series meta-learner $\Theta$, and general meta-learner $\Phi$. The meta-learner $\mathcal{L}$ has three main inputs; the performance tensor $\mathsf{P}$, the meta-features tensor $\mathsf{F}$, and the loss function. In the offline training of the meta-learner $\mathcal{L}$, it learns two components $\Theta$ and $\Phi$, where the former captures the temporal relationship between the meta-features of the consecutive time windows within the same dataset and the evolution of the performances of the models on these windows (i.e., the best candidate models on those time-windows). On the other hand, the general meta-learner $\Phi$ predicts the best model for each task (window) without taking into account the temporal relationship among different time windows within the same dataset.

**Rationale for Having both General and Time-series Meta-learners:** The rationale of having both meta-learners is the fact that the temporal dependency among different time windows depends on the dataset type. Some datasets have strong temporal dependency which would be predicted efficiently by the time-series meta-learner $\Theta$ while others datasets would have weak temporal dependence among different windows performances in which the general meta-learner $\Phi$ is expected to perform better. We show the results of such different datasets for our two testbeds in Table 12-13.

---

[3]Note that we do feature-embedding (PCA), as shown in Figure 1, to get the final meta-features tensor $\boldsymbol{F}_i$.

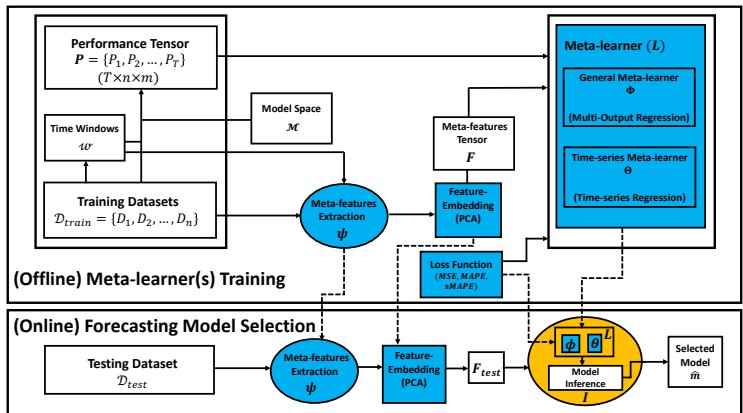

Figure 1: An overview of AUTOFORECAST; components that transfer from offline to online (model selection) phase shown in blue. Given the two main inputs; the performance tensor **P** and the meta-features tensor **F**, the meta-learner $\mathcal{L}$ learns two main components; general meta-learner ($\mathbf{\Phi}$) and time-series meta-learner ($\mathbf{\Theta}$).

**General Meta-learner $\mathbf{\Phi}$:** We propose *multi-output regression model* for training our general meta-learner $\mathbf{\Phi}$ in Figure 1. From running all the models in $\mathcal{M}$ on different time windows $w_t$ with $t \in \{1, \ldots, T\}$ for all datasets in the meta-train database $\mathcal{D}_{train}$, we collect a set of $N = T \times n$ distinct training samples $(\boldsymbol{F}_t^i, \boldsymbol{p}_t^i)$, with $t \in [1, T]$ and $i \in [1, n]$. Recall that $\boldsymbol{F}_t^i \in \mathbb{R}^{d \times v_i}$ is the meta-feature matrix of the window $w_t$ of dataset $\boldsymbol{D}_i \in \mathcal{D}_{train}$ and that $\boldsymbol{p}_t^i \in \mathbb{R}^{|\mathcal{M}|}$ (performance vector on window $w_t$ of $\boldsymbol{D}_i$). Thus, the multi-output regression model is given by

$$\hat{\boldsymbol{p}}_t^i = \mathbf{\Phi}\left(\boldsymbol{F}_t^i, \boldsymbol{\beta}\right); t \in [1, T], i \in [1, n], \tag{3}$$

where $\mathbf{\Phi}$ denotes the regression function (e.g., linear, NN) and $\boldsymbol{\beta}$ are the unknown regression parameters. Thus, the general meta-learner's objective, denoted by the loss function $L_{\mathbf{\Phi}}$, is given by

$$L_{\mathbf{\Phi}} = \sum_{t=1}^{T} \sum_{i=1}^{n} L(\hat{\boldsymbol{p}}_t^i, \boldsymbol{p}_t^i), \tag{4}$$

where $L$ is the loss metric (e.g., MSE, MAPE, etc). Therefore, $\mathbf{\Phi}$ learns the mapping between the meta-features of a time window in a dataset and the corresponding best model in the model space.

**LSTM-based Time-series Meta-learner $\mathbf{\Theta}$:** The goal of the time-series meta-learner $\mathbf{\Theta}$ (in Figure 1) is to learn how the models' performances evolve with the time-series meta-feature matrices over time. For such purpose, we propose *time-series multi-regression model* to learn such performance evolution. For any dataset $\boldsymbol{D}_i$, given the time-series meta-feature matrices $\boldsymbol{F}_1^i, \ldots, \boldsymbol{F}_{t-1}^i, \boldsymbol{F}_t^i$ and the history of the performance vectors $\boldsymbol{p}_1^i, \ldots, \boldsymbol{p}_{t-1}^i$, we aim to predict performance vector $\boldsymbol{p}_t^i$ of current time window $w_t$. The time-series regression equation would be

$$\hat{\boldsymbol{p}}_t^i = \mathbf{\Theta}\left(\boldsymbol{F}_1^i, \boldsymbol{F}_2^i, \ldots, \boldsymbol{F}_{t-1}^i, \boldsymbol{F}_t^i, \boldsymbol{p}_1^i, \boldsymbol{p}_2^i, \ldots, \boldsymbol{p}_{t-1}^i\right), i \in [1, n], t \in [1, T],$$

where $\mathbf{\Theta}$ denotes the time-series regression function, and $\hat{\boldsymbol{p}}_t^i$ is the predicted performance vector.

We adapt long-short term memory (LSTM) inputs for our time-series meta-learner $\mathbf{\Theta}$. We denote $\boldsymbol{X}_t$ as the input at the time window $w_t$ which is given by $\boldsymbol{X}_t = \left[\boldsymbol{F}_1^i, \boldsymbol{p}_1^i, \boldsymbol{F}_2^i, \boldsymbol{p}_2^i, \cdots, \boldsymbol{F}_{t-1}^i, \boldsymbol{p}_{t-1}^i, \boldsymbol{F}_t^i\right]$. The predicted LSTM's output denoted by $\hat{\boldsymbol{p}}_t^i$ is a function of $\boldsymbol{X}_t$; detailed equations of such relation between $\hat{\boldsymbol{p}}_t^i$ and $\boldsymbol{X}_t$ in the LSTM are provided in Appendix D. During training, $\mathbf{\Theta}$ learns the parameters $\boldsymbol{W}_f, \boldsymbol{b}_f, \boldsymbol{W}_l, \boldsymbol{b}_l, \boldsymbol{W}_o, \boldsymbol{b}_o, \boldsymbol{W}_u, \boldsymbol{b}_u, \boldsymbol{W}_v, \boldsymbol{b}_v$ which are the weights and biases of the forget, input, and output layers and cell updates, respectively. Thus, the objective of the LSTM time-series meta-learner $\mathbf{\Theta}$, denoted by $L_{\mathbf{\Theta}}$, would be given by

$$L_{\mathbf{\Theta}} = \sum_{t=1}^{T} \sum_{i=1}^{n} L(\hat{\boldsymbol{p}}_t^i, \boldsymbol{p}_t^i). \tag{5}$$

We emphasize that $\mathbf{\Theta}$ learns temporal relationships between the history of the meta-features and performance vectors over time windows and the corresponding current best model in model space.

**Meta-learner Objective**: Having established the two main components — the general meta-learner $\mathbf{\Phi}$ and the time-series meta-learner $\mathbf{\Theta}$) — we now define the objective of our meta-learner $\mathcal{L}$, which is given by a linear combination of the two components as follows:

$$\min_{\boldsymbol{\beta}, \boldsymbol{W}_f, \boldsymbol{W}_l, \boldsymbol{W}_o, \boldsymbol{b}_f, \boldsymbol{b}_l, \boldsymbol{b}_o, \boldsymbol{W}_u, \boldsymbol{b}_u, \boldsymbol{W}_v, \boldsymbol{b}_t} a L_{\mathbf{\Phi}}(\mathbf{F}, \mathbf{P}) + b L_{\mathbf{\Theta}}(\mathbf{F}, \mathbf{P}). \tag{6}$$

The meta-learner $\mathcal{L}$ learns jointly general meta-learner $\mathbf{\Phi}$ (equation 4) and time-series meta-learner $\mathbf{\Theta}$ (equation 5) given meta-learner inputs, the performance tensor $\mathbf{P}$ and the meta-features tensor $\mathbf{F}$. By definition, this meta-learner $\mathcal{L}$ optimizes the loss over all datasets and all time windows.

## 4.2 Online Inference And Model Selection

In the online mode of AUTOFORECAST, we aim to make use of the trained meta-leaner $\mathcal{L}$ to quickly infer the best model for the current task. Given a new time-series dataset $\boldsymbol{D}_{test}$, AUTOFORECAST first computes the corresponding meta-features tensor by $\hat{\mathbf{F}}_{test} = \psi(\boldsymbol{D}_{test})$. Those time-series meta-features are then embedded (using PCA) to obtain the final meta-features tensor $\boldsymbol{F}_{test}$. Then, in the model inference, as shown in Figure 1, the model set performances are predicted for each available model in $\mathcal{M}$. The model $\hat{M}_t$ with the lowest predicted error score by $\mathcal{L}$ on the time window $w_t$ of $\boldsymbol{D}_{test}$ is chosen as the selected model for that window $w_t$. Now, we explain such model selection process for each time window across the time windows $w_0, w_1, \ldots, w_T$ of $\boldsymbol{D}_{test}$ as follows. For the first window ($w_0$), the inference is given by $\hat{M}_0 \in \arg\min_{\bar{M} \in \mathcal{M}} \mathcal{L}(\boldsymbol{F}_0^{test})$. For any other window $w_t$ (with $t > 0$), the time-series meta-learner $\mathbf{\Theta}$ inference depends on the history of the meta-features and the history of the models' performances as follows $\hat{M}_t^{\mathbf{\Theta}} \in \arg\min_{\bar{M} \in \mathcal{M}} \mathbf{\Theta}(\boldsymbol{F}_0^{test}, \ldots, \boldsymbol{F}_{t-1}^{test}, \boldsymbol{F}_t^{test}, \hat{\boldsymbol{p}}_0^{test}, \ldots, \hat{\boldsymbol{p}}_{t-1}^{test})$. On the other hand, the general meta-learner $\mathbf{\Phi}$ inference depends on the predicted (regression) output on the meta-features of current time window where $\hat{M}_t^{\mathbf{\Phi}} \in \arg\min_{\bar{M} \in \mathcal{M}} \mathbf{\Phi}(\boldsymbol{F}_t^{test})$. Thus, the final selected model is given by

$$\hat{M}_t \in \underset{\bar{M} \in \{\hat{M}_t^{\mathbf{\Phi}}, \hat{M}_t^{\mathbf{\Theta}}\}}{\arg\min} \hat{\boldsymbol{p}}_t^{test}(\bar{M}). \tag{7}$$

We emphasize that tie between models can happen in online inference (i.e., two or more models can have an identical predicted performance). We built upon the several tie breaking techniques that have been examined in the literature Kalousis (2002); Brazdil et al. (2008), but usually such a choice does not have a strong influence on the performance of a meta-learning system. For making the decision between the model selected by the general meta-learner $\mathbf{\Phi}$ and that selected by the time-series meta-learner $\mathbf{\Theta}$, we choose the model with the least predicted error score (illustrated in equation 7). We present the time complexity analysis of AUTOFORECAST's inference in Appendix H.

## 5 Experiments

We evaluate AUTOFORECAST by designing experiments to answer the following research questions:

- Does employing AUTOFORECAST for time-series forecasting model selection yield improved performance, as compared to no model selection, as well as other selection techniques (such as meta-learners adapted from the AutoML domain)?
- How much reduction in inference time does AUTOFORECAST give over the naïve method?
- How does performance change with different datasets with different temporal dependence?

## 5.1 Experimental Setup

**Models and Performance Collection:** By pairing seven SOTA time-series forecasting algorithms (which are DeepAR Salinas et al. (2020), Deep Factors Wang et al. (2019), Prophet Taylor & Letham (2018), Seasonal Naive Montero-Manso et al. (2020), Gaussian Process Yan et al. (2009), Vector Auto Regression Lewis & Reinsel (1985), and Random Forest Regressor Liaw et al. (2002)) and their corresponding hyperparameters, and using different data representation methods, we compose a model set $\mathcal{M}$ with 322 unique models (see Table 9 for the complete list). For our testbeds, we first generate the performance tensor $\mathbf{P}$, by evaluating the models from $\mathcal{M}$ against the benchmark datasets in each testbed. For consistency, all models are built using the GluonTS Alexandrov et al.

Table 1: Time-series dataset corpus statistics and properties.

| # time-series datasets $|\mathcal{D}|$ | # overall time-series (across all datasets) | # of multivariate time-series datasets | # of univariate time-series datasets |
|---|---|---|---|
| **348** | 625 | 40 | 308 |

(2020), Scikit-learn Pedregosa et al. (2011), and Statsmodels Seabold & Perktold (2010) Python libraries on an Intel i7 @2.60 GHz, 16GB RAM, 8-core workstation.

**Meta-Train Testbed:** Meta-learning works if the new task can leverage prior knowledge. We thus create two testbeds with 348 public forecasting datasets. In particular, we collect 308 univariate time-series datasets for the first testbed (Table 10 in Appendix G) and 40 multivariate datasets with 317 time-series for the second testbed (Table 11 in Appendix G). In total, we have 625 time-series in our testbeds. We refer to Table 1 for a summary of the statistics of the corpus of time-series data we used. For each dataset in the testbeds, we use different time windows selected randomly from the dataset, where each time window has a length of 16 (i.e., $|w_t| = 16 \; \forall t \in \{1, \ldots, T\}$). We emphasize that our approach has no restriction on the length of the time window.

**Evaluation:** For evaluating AUTOFORECAST, we split each testbed into 5 folds for cross-validation. For each fold, after training the meta-learning approach, we use it to infer the best forecasting model for the new unseen test datasets. Finally, we take the average performance of these five folds. We mainly compare the Hit-at-$k$ accuracy (which indicates whether the selected model is within the actual top-$k$ models), of AUTOFORECAST against different meta-learners baselines. We also compare the performance of selected model by AUTOFORECAST against the performance of the selected model by other meta-learners. This is measured by the mean square error (MSE) and the average rank (for each testbed, the meta-learners are first ranked by the corresponding forecasting MSE under the selected model for each dataset and then the rank is averaged across all datasets).

**Time-series Meta-learner Setup:** For our time-series meta-learner $\Theta$ explained in Section 4, we used LSTM with 4 layers where each layer has 50 units. The training was with 50 epochs with the Adam optimizer with a batch size of 25 and dropout rate of 0.2 to prevent over-fitting. The evaluation of the effect of such parameters on $\Theta$'s performance is shown in Appendix J.3.

## 5.2 BASELINES

We adapt leading methods from algorithm selection, and include additional baselines by creating a variant of the proposed AUTOFORECAST (marked with TSL). All the baselines can be organized into the following categories (the detailed description of each baseline is given in Appendix I).

**No model selection**: This category always employs either the same single model or the ensemble of all the models, e.g., **Random Forest (RF)** Liaw et al. (2002).

**Simple meta-learners**: Meta-learners in this category pick the generally well-performing forecasting model, globally or locally (via clustering or KNN): **Global Best (GB)**, **ISAC** Kadioglu et al. (2010), and **ARGOSMART (AS)** Nikolić et al. (2013).

**Optimization-based meta-learners**: These Meta-learners learn task similarities via optimizing performance estimates on meta-features: **Multi-layer Perceptron (MLP)**, and **AUTOFORECAST-TSL**; a variant in which meta-learner $\mathcal{L}$ consists only of time-series learner $\Theta$.

## 5.3 RESULTS

Now, we show our results to answer the aforementioned evaluation questions.

### 5.3.1 UNIVARIATE TESTBED RESULTS

To investigate the impact of the train/test similarity on meta-learning performance, we build the univariate testbed that consists of 308 datasets (Appendix G Table 10) with diverse datastes.

**Superiority of** AUTOFORECAST **compared to all baseline methods w.r.t. the Hit-at-$k$, average rank, and MSE:** The different results are provided in Table 2-4 where the best result for every testbed is highlighted in bold. We observe that AUTOFORECAST outperforms previous SOTA meta-learning methods adapted to our problem. For example, AUTOFORECAST has 79.20%, 171.86%, 423.17%, 375.61%, and 51.59% higher Hit-at-10 accuracy than GB, AS , ISAC, MLP, and AUTOFORECAST-TSL, respectively. Full evaluation of univariate testbed is in Appendix J.

Table 2: Hit-at-$k$ Accuracy (the higher the better) comparison of AUTOFORECAST against the different baseline meta-learners for both univariate and multivariate testbeds. AUTOFORECAST outperforms all baselines for both testbeds.

| Dataset Testbed | k | Global Best | AS | ISAC | MLP | AUTOFORECAST-TSL | AUTOFORECAST |
|---|---|---|---|---|---|---|---|
| **Univariate** | 1 | 2.46 | 2.15 | 0.82 | 0.62 | 2.67 | **3.95** |
| | 5 | 7.18 | 4.92 | 2.67 | 1.13 | 9.04 | **14.57** |
| | 10 | 11.97 | 7.89 | 4.10 | 4.51 | 14.15 | **21.45** |
| | 50 | 37.40 | 28.00 | 11.45 | 22.25 | 35.28 | **52.05** |
| **Multivariate** | 1 | **6.78** | 2.26 | 4.19 | 0.43 | 5.16 | 5.87 |
| | 5 | 12.18 | 4.73 | 5.69 | 1.51 | 9.03 | **13.86** |
| | 10 | 16.21 | 9.03 | 7.31 | 4.06 | 11.39 | **20.91** |
| | 50 | 41.72 | 24.73 | 14.64 | 20.86 | 35.06 | **51.67** |

Table 3: Average rank (the lower the better) comparison of AUTOFORECAST against the different baseline meta-learners for both testbeds. AUTOFORECAST outperforms all baselines.

| Dataset Testbed | Global Best | AS | ISAC | MLP | AUTOFORECAST-TSL | AUTOFORECAST |
|---|---|---|---|---|---|---|
| **Univariate** | 2.5161 | 2.7965 | 2.9096 | 3.7072 | 2.5202 | **2.0571** |
| **Multivariate** | 2.3191 | 3.0851 | 2.3191 | 3.8723 | 2.3404 | **1.3191** |

Table 4: Results for one-step ahead forecasting (MSE) for both testbeds. The selected model by AUTOFORECAST yields better performance (i.e., lower MSE) compared to baseline meta-learners.

| Dataset Testbed | Global Best | AS | ISAC | MLP | AUTOFORECAST-TSL | AUTOFORECAST |
|---|---|---|---|---|---|---|
| **Univariate** | $0.0065 \pm 0.0199$ | $0.0158 \pm 0.0556$ | $0.0071 \pm 0.0145$ | $0.0351 \pm 0.1186$ | $0.00463 \pm 0.0138$ | $\mathbf{0.00256 \pm 0.0090}$ |
| **Multivariate** | $0.0046 \pm 0.0099$ | $0.0139 \pm 0.0563$ | $0.0046 \pm 0.0099$ | $0.0121 \pm 0.2462$ | $0.00541 \pm 0.0186$ | $\mathbf{0.00124 \pm 0.0051}$ |

**Statistical Significance of** AUTOFORECAST**:** To compare two methods statistically, we use the pairwise Wilcoxon rank test on performances (i.e., MSE of selected models) across datasets (significance level $p < 0.05$). Table 5 shows that AUTOFORECAST is significantly better than most of the baseline meta-learners, i.e., including GB ($9.07 \times 10^{-5}$), AS ($1.07 \times 10^{-37}$) and AUTOFORECAST-TSL ($8.16 \times 10^{-15}$). Appendix J.2 has full statistical significance tests.

**Meta-learners perform better than methods without model selection:** As Table 14 (Appendix J) shows, the meta-learners outperforms almost all models with no model selection. In particular, three meta-learners (Global Best, ISAC, AUTOFORECAST) significantly outperform the baseline time-series forecasting models. For instance, AUTOFORECAST respectively has 92.58%, 84.39%, 88.20%, 87.14%, 83.48%, 98.45%, and 95.75% lower MSE over Seasonal Naive, DeepAR, Deep Factors, Random Forest, Prophet, Gaussian Process, and VAR, respectively. These results signify the benefits of using meta-learning for model selection, specifically using AUTOFORECAST.

**Optimization-based meta learners generally perform better than simple meta learners:** Two of the top-3 meta learners by average rank and MSE (AUTOFORECAST and AUTOFORECAST-TSL) are all optimization-based and significantly outperform simple meta-learners such as ISAC and AS as shown in Table 2 and Table 4. The interpretation is that simple meta-learners weigh meta-features equally for task similarity, whereas optimization-based methods learn which meta-features matter (e.g., time-series regression on meta-features in AUTOFORECAST-TSL), leading to better results.

**Dataset-wise Performance:** We present the detailed performances for each dataset by comparing AUTOFORECAST with all baseline methods in Table 12. It is noted these results are averaged across the different time windows for each dataset. The results shows that AUTOFORECAST achieves the best MSE and average rank among all meta-learners. We note that AUTOFORECAST and AUTOFORECAST-TSL have same performance for datasets with higher temporal dependency.

### 5.3.2 MULTIVARIATE TESTBED RESULTS

In this testbed, we simulate the case when there are similar meta-train tasks to the test task, where we choose variables from the multivariate for training and the rest in testing. We build the multivariate testbed that consists of 40 datasets (Table 11 in Appendix G).

**For the Multivariate testbed,** AUTOFORECAST **still outperforms all baseline methods w.r.t. average rank, MSE, and Hit-at-$k$ accuracy** as shown in Tables 2-4. Moreover, Figure 8 (Appendix J) shows that for the pool of multivariate datasets (across all time windows), AUTOFORECAST gives a gain of 2X in selecting better models as compared to other meta-learning baselines. **Dataset-wise** performance for the datasets in the multivariate testbed is shown in Table 13 in Appendix J. AUTOFORECAST has the lowest average MSE on most of the multivariate datasets.

Table 5: Pairwise statistical test results between AUTOFORECAST and baselines by Wilcoxon signed rank test. Statistically better method ($p = 0.05$) shown in **bold** (both marked bold if no significance). In (left), Univariate testbed is shown. In (right), Multivariate testbed is shown. For both testbeds, AUTOFORECAST is statistically better than most of the baseline meta-learners.

| Ours | Baseline | p-value |
|---|---|---|
| **AUTOFORECAST** | **GB** | $9.0712 \times 10^{-5}$ |
| **AUTOFORECAST** | **AS** | $1.0726 \times 10^{-37}$ |
| **AUTOFORECAST** | **ISAC** | 0.1349 |
| **AUTOFORECAST** | **MLP** | 0.0657 |
| **AUTOFORECAST** | AUTOFORECAST-TSL | $8.1683 \times 10^{-15}$ |

| Ours | Baseline | p-value |
|---|---|---|
| **AUTOFORECAST** | **GB** | 1.0 |
| **AUTOFORECAST** | **AS** | $3.9399 \times 10^{-7}$ |
| **AUTOFORECAST** | **ISAC** | 0.8240 |
| **AUTOFORECAST** | **MLP** | 0.0004 |
| **AUTOFORECAST** | AUTOFORECAST-TSL | 0.0025 |

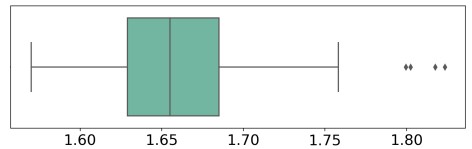

Figure 2: Boxplot of AUTOFORECAST inference time for univariate testbed (in sec.). AUTOFORECAST takes less than 1.7 sec for most datasets (median = 1.65).

Figure 3: The inference time reduction of AUTOFORECAST over the naïve approach. AUTOFORECAST gives a median reduction of 42X over naïve approach for both testbeds.

Table 6: Inference runtime performance (in seconds) for both univariate and multivariate testbeds.

| Dataset Testbed | Global Best | AS | ISAC | MLP | AUTOFORECAST-TSL | AUTOFORECAST |
|---|---|---|---|---|---|---|
| **Univariate** | $0.6259 \pm 0.0964$ | $0.8537 \pm 0.1438$ | $10.2480 \pm 2.7182$ | $1.2745 \pm 0.5198$ | $0.7962 \pm 0.0436$ | $1.6508 \pm 0.0401$ |
| **Multivariate** | $0.4151 \pm 0.0403$ | $1.3055 \pm 0.2610$ | $7.037 \pm 1.6239$ | $1.1461 \pm 0.2176$ | $0.682 \pm 0.0372$ | $1.1309 \pm 0.1257$ |

**Statistical Significance of** AUTOFORECAST: Table 5 shows that for the Multivariate testbed, AUTOFORECAST is also significantly better than most of the baseline meta-learners, i.e., including AS ($3.94 \times 10^{-7}$), MLP (0.0004) and AUTOFORECAST-TSL (0.0025) and has no significance from GB and ISAC. For the full statistical significance results for all pairs see Table 15 in Appendix J.2.

### 5.3.3 RUNTIME ANALYSIS

**Inference run time statistics of** AUTOFORECAST: Figure 2 shows that AUTOFORECAST (meta-feature generation and model selection) takes less than 1.7 seconds on most time series datasets. Moreover, Figure 3 shows that AUTOFORECAST has significant reduction in inference time compared to the naïve approach, median is 42X on both testbeds (see also Appendix J.5).

**Comparing** AUTOFORECAST **with baselines:** In terms of inference, Table 6 shows that most of the meta-learners are fast taking only a few seconds to infer the best forecasting model. Finally, we compare the training cost of AUTOFORECAST against baseline meta-learners. Table 16 shows that AUTOFORECAST has comparable computational training cost. While the training process is offline and done only once and hence, not as important as the inference time cost, this experiment reassures us that our better model selection performance does not entail a prohibitive training cost.

## 6 CONCLUSION

We introduced a meta-learning approach to automate the process of time-series forecasting by automatically inferring the best time-series model on an unseen dataset, without needing exhaustive evaluation of all existing models on this dataset. The problem arises because there are many possible forecasting models with their associated hyperparameters, and different choices are optimal for different datasets. Further, even within one non-stationary dataset, different models are appropriate for different time windows. Our proposed solution AUTOFORECAST is a meta-learner, trained on an extensive pool of historical time-series forecasting datasets and models. To effectively capture task similarity, we designed novel problem-specific meta-features. Extensive experiments on two large testbeds showed that AUTOFORECAST significantly improves time-series forecasting model selection over directly using some of the most popular models as well as several SOTA meta-learners. We showed that AUTOFORECAST gives a significant improvement in the inference time compared to naïve approaches. We release the benchmark data for the community to contribute new datasets and models to stimulate further advances on automating time-series forecasting.

**Reproducibility Statement:** To further research into the important problem introduced in our work, we have publicly released our benchmark data to to enable others reproduce our work. In particular, we are publicly releasing, with this submission, our meta-learning database corpus of 348 datasets, containing 625 time series in all, performances of the 322 forecasting models, and meta-features for the datasets. This resource will hopefully encourage the community to standardize efforts at benchmarking time series forecasting model selection. We also encourage the community to expand this resource by contributing their new datasets and models. The anonymized website with our database is: https://drive.google.com/drive/folders/1K1w1Ida5Cr15b5Fhidax-i-fNpWZjvet. The details of each dataset in the two testbeds, univariate (308 datasets) and multivariate (40 datasets) are explained in Appendix G (Table 10 and Table 11) and the complete list of meta-features is presented in Appendix E (Table 8).

The performance collection is difficult and time-consuming to obtain, yet fundamental for researchers to begin studying the important problem formulated in our work, and developing better approaches to solve it. Here, we give the result of that time consuming process of collecting the performance of each of the 322 forecasting models on all the datasets (in the above link). In particular, we collect this for the performance metrics of MSE, training time, and inference time. This serves as the training data and ground truth evaluation data for AUTOFORECAST and all other competing protocols. We refer the reader to Appendix F in which we provide complete description of the algorithms, libraries, and hyper-parameter values used for this collection process.

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

APPENDIX

## A    VALIDATION OF NO-SINGLE-MODEL HYPOTHESIS ON OUR TESTBEDS

Figure 4 shows a validation of the hypothesis that there is no unique single model that works well on all datasets, where we show a histogram of the best models probability distribution across the datasets of the training testbed where different datasets have different (best) models. That motivates the need for our framework, AUTOFORECAST, for automating model selection via meta-learning.

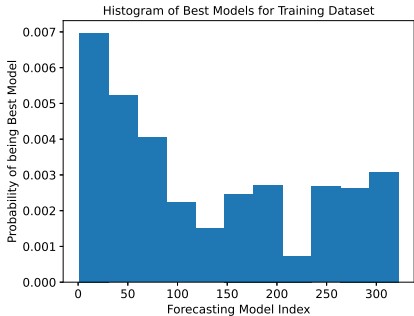

Figure 4: A histogram of the best forecasting model's probability distribution across the datasets of the training testbed. We observe that different datasets have different best models and no single model triumphs across all datasets. This motivates our automatic model selection problem.

## B    VARIATION OF BEST MODEL ACROSS TIME WINDOWS

Figure 5 validate the no-free lunch theorem across different datasets (vertically) and across different time windows within the same dataset (Horizontally). Figure 6 shows the aggregate statistics on all datasets of the univariate testbed. This contradicts the claims that one forecasting algorithm can work best for different datasets and motivates the need for an effective approach for learning such both dimensions for selecting the best model for new tasks, which we propose in our current work.

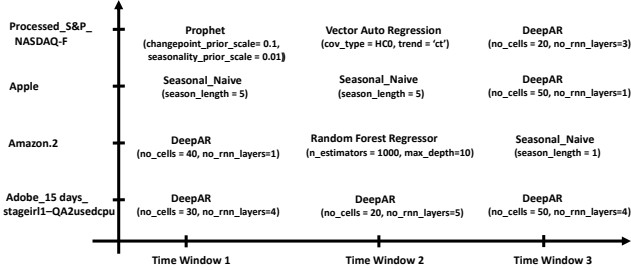

Figure 5: The best forecasting model for each time window (across three consecutive time windows) for different datasets. We observe that different time windows have different (best) models.

## C    SUMMARY OF NOTATION

We summarize the notation used in this paper in Table 7.

### C.1    WINDOW NOTATION IN AUTOFORECAST

We now reemphasize the notation of time window we used in Section 4. In AUTOFORECAST, the time window represents a sequence of time observations in the time series. In particular, $w_t$ denotes the time window index, and $|w_t|$ is the length of that time window (e.g., $|w_{10}| = 16$ means that the 10th time window of the dataset has a length of 16 observations). Notice that for each time window $w_t$, we have the pair $(x_{t-|w_t|:t-1}, y_{t:t+H})$, where the different models are trained on $x_{t-|w_t|:t-1}$ and

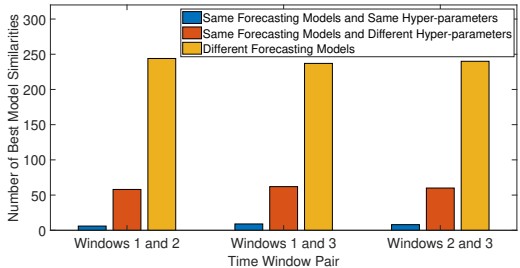

Figure 6: The aggregate statistics for similarity in the best forecasting model across three consecutive time windows for univariate testbed. Most different time windows have different (best) models.

Table 7: A summary of our notation.

| Symbol | Description |
| --- | --- |
| $\mathcal{D}_{train}$ | Meta-train database |
| $n$ | Meta-train database size |
| $n_i$ | No. of observations in $\boldsymbol{D}_i$ |
| $v_i$ | No. of variables in $\boldsymbol{D}_i$ |
| $\mathcal{M}$ | Model space |
| $M_j$ | A forecasting model |
| $m$ | Size of the model space |
| $\boldsymbol{h}_i$ | Hyperparameter vector of $a_i$ |
| $T$ | Number of time windows |
| $|w_i|$ | Window length |
| $\mathbf{P}$ | Performance tensor |
| $\boldsymbol{P}_k$ | Performance matrix of window $w_k$ |
| $p_k^{i,j}$ | Model $M_j$'s performance on time window $w_k$ of $\boldsymbol{D}_i$ |
| $\boldsymbol{p}_k^i$ | Models performances vector on time window $w_k$ for $\boldsymbol{D}_i$ |
| $g_i(\cdot)$ | Data representation |
| $\mathbf{F}_i$ | Meta-features tensor of $\boldsymbol{D}_i$ |
| $\boldsymbol{F}_k^i$ | Meta-feature matrix of window $w_k$ in $\boldsymbol{D}_i$ |
| $\psi(\cdot)$ | Meta-features extraction module |
| $d$ | Number of meta-features |
| $\mathcal{L}$ | Meta-learner |
| $\Phi$ | General Meta-learner |
| $\Theta$ | Time-series Meta-learner |
| $\hat{M}_t^{\Theta}, \hat{M}_t^{\Phi}$ | Selected model for time window $w_t$ by $\Theta$ and $\Phi$, respectively |
| $\hat{M}_t$ | Selected model for window $w_t$ in inference by AUTOFORECAST |

evaluated on $y_{t:t+H}$, where $H$ is the forecasting horizon. For example, for single-step forecasting on a window $w_t$ with length 16, we would train the forecasting model on the first 15 observation points of the window and do forecasting on the 16th observation. Second, we emphasize that we can train a forecasting model on windows with any length, but the forecasting horizon may be limited by the forecasting model parameters.

## D  ADDITIONAL DETAILS OF TIME-SERIES META-LEARNER

We now detail the equations of our LSTM-based time-series meta-learner $\Theta$. The LSTM cell at time $t$ has two recurrent features, denoted by $\boldsymbol{h}_t^i$ and $\boldsymbol{c}_t^i$, called the hidden state and the cell state, respectively. The LSTM cell consists of three layers (forget gate layer, input gate layer, and output gate layer). The activation of those layers is given by

$$\boldsymbol{f}_t^i = \sigma\left(\boldsymbol{W}_f \cdot [\boldsymbol{h}_{t-1}^i, \boldsymbol{X}_t] + \boldsymbol{b}_f\right), \tag{8}$$

$$\boldsymbol{l}_t^i = \sigma\left(\boldsymbol{W}_l \cdot [\boldsymbol{h}_{t-1}^i, \boldsymbol{X}_t] + \boldsymbol{b}_l\right), \tag{9}$$

$$\boldsymbol{o}_t^i = \sigma\left(\boldsymbol{W}_o \cdot [\boldsymbol{h}_{t-1}^i, \boldsymbol{X}_t] + \boldsymbol{b}_o\right). \tag{10}$$

where $\boldsymbol{W}_f, \boldsymbol{W}_l, \boldsymbol{W}_o$ and $\boldsymbol{b}_f, \boldsymbol{b}_l, \boldsymbol{b}_o \in \mathbb{R}^m$ denote the weights matrices and the biases of the three layers, respectively. These are the parameters to be learned during the training of the time-series meta-learner. The forget gate $\boldsymbol{f}_t^i$ controls how much of the current cell state we should forget, the input gate $\boldsymbol{l}_t^i$ controls how much of the cell update is added to the cell state, and the output gate $\boldsymbol{o}^i$ controls how much of the modified cell state should leave the cell and become the next hidden state. Moreover, the cell update $\mathbf{u}_t^i$ is constructed with a $\tanh$ activation function as follows.

$$\mathbf{u}_t^i = \tanh\left(\boldsymbol{W}_u \cdot [\boldsymbol{h}_{t-1}^i, \boldsymbol{X}_t] + \boldsymbol{b}_u\right), \tag{11}$$

where $\boldsymbol{W}_u$ and $\boldsymbol{b}_u \in \mathbb{R}^m$ are further weight parameters to be learned. Thus, the new cell and hidden states at time $t$ are given by

$$\boldsymbol{c}_t^i = \boldsymbol{f}_t^i \cdot \boldsymbol{c}_{t-1}^i + \boldsymbol{l}_t^i \cdot \mathbf{u}_t^i \tag{12}$$

$$\hat{\boldsymbol{h}}_t^i = \boldsymbol{o}_t^i \cdot \tanh(\boldsymbol{c}_t^i) \tag{13}$$

Finally, the output equations of the LSTM cell are given by

$$\mathbf{V}_t^i = \boldsymbol{W}_v \hat{\boldsymbol{h}}_t^i + \boldsymbol{b}_v \tag{14}$$

$$\hat{\boldsymbol{p}}_t^i = \text{Softmax}(\mathbf{V}_t^i), \tag{15}$$

where $\boldsymbol{W}_v$ and $\boldsymbol{b}_v \in \mathbb{R}^m$ are weight parameters to be learned. This gives the relationship between input $\boldsymbol{X}_t$ and predicted performances output $\hat{\boldsymbol{p}}_t^i$. For ease of notation, we considered a single layer in the above analysis . However, we have used multiple layers with several units (see Section 5).

# E  TIME-SERIES META FEATURES

There are prior works that generated standard time-series features Franceschi et al. (2019), tsfresh Christ et al. (2018) (that we used for generating part of our meta-features). We now provide details of our meta-features and their different categories.

## E.1  META-FEATURES CATEGORIES:

For each dataset, we generate a meta-feature vector that consists of more than 800 meta-features. This meta-features vector includes different components that capture various properties of the time-series dataset, *e.g.*, the statistical features (number of crossings, count of observations above/below the mean value, quantiles, etc.), the data trend (linear, non-linear, dynamics, etc.), data interdependence (lag autocorrelation, difference features, etc.), information-theoretic features that are typically based on entropy measures, and the frequency (FFT, wavelet, etc.). Our meta-features vector also includes landmarker features, which are problem-specific, and aim to capture the unique characteristics of a dataset. The idea is to apply a few of the fast, easy-to-construct time-series forecasting models on a dataset and extract features from (i) the structure of the estimated forecasting model, and (ii) its output performance scores.

**Multivariate meta-features:** For this time-series dataset type, we have two types (explained in Appendix G). For both types, we generate a single meta-features vector of the multivariate dataset by averaging the meta-features vectors of all variables within the dataset.

## E.2  COMPLETE LIST OF FEATURES

We summarize the meta-features used by AUTOFORECAST in Table 8. When applicable, we provide the formula for computing the meta-feature(s) and corresponding variants we used for generating the meta-features, and the corresponding number of features. Some are based on Vanschoren (2018). Specifically, our meta-features can be categorized into (1) simple features, (2) statistical features, (3) information-theoretic features, (4) Spectral features, and (5) landmarker features. Broadly speaking, the statistical features captures statistical properties of the underlying data distributions; e.g., min, max, variance, skewness, covariance, etc. of the features and feature combinations. The information-theoretic features capture information-theoretic underlying characteristics in the time-series; e.g., entropy, trend, non-linearity, change statistics, etc. Most of those meta-features have been commonly used in the AutoML literature Vanschoren (2018). To the best of our knowledge, we emphasize that our landmarker meta-features (detailed below) are novel and that some components of the spectral meta-features have not been used in any related work.

## E.3  LANDMARKER META-FEATURE GENERATION

In addition to simple, statistical, and information-theoretic meta-features, we use three time-series forecasting landmarker algorithms for computing forecasting-specific landmarker meta-features, Auto Regression Lewis & Reinsel (1985), Random Forest Liaw et al. (2002), and Bayesian Ridge Regression Shi et al. (2016) to capture landmarker characteristics of a time-series dataset.

We now provide a quick overview of each algorithm and then discuss how we are using them for building meta-features. The algorithms are executed with the default parameter.

**Auto Regression Lewis & Reinsel (1985)**: In this landamarker feature, we fits the unconditional maximum likelihood of an autoregressive process. The $k$ parameter represents the maximum lag of the process

$$X_t = \varphi_0 + \sum_{i=1}^{k} \varphi_i X_{t-i} + \varepsilon_t$$

Then, we extract the coefficients $\varphi_i$ whose index $i \in \{0, \cdots, k\}$.

**Random Forest Liaw et al. (2002)**: is a tree-based ensemble method that builds a collection of base trees using the subsampled unlabeled data, splitting on (randomly selected) features as nodes. Random Forest grows internal nodes until the terminal leaves contain only one sample or the predefined max depth is reached.

For Random Forest, we use the balance of base trees (i.e., depth of trees and number of leaves per tree) and additional information (e.g., feature importance of each base tree). It is noted that feature importance information is available for each base tree—we therefore analyze the statistic of mean and max of base tree feature importance. The following information of base trees are used:

- Tree depth: min, max, mean, std, skewness, and kurtosis
- Number of leaves: min, max, mean, std, skewness, and kurtosis
- Mean of base tree feature importance: min, max, mean, std, skewness, and kurtosis
- Max of base tree feature importance: min, max, mean, std, skewness, and kurtosis
- Out-of-bag estimate score.

**Bayesian Ridge Regression Shi et al. (2016)**: it estimates a Gaussian probabilistic model of the regression problem using Bayesian regression. The prior for the coefficient is given by a spherical Gaussian distribution.

For Bayesian Ridge Regression, we use the following information of the probabilistic model (fitted using the dataset) as landmarker features:

- Mean of the distribution
- Log marginal Likelihood score
- The precision of the estimated weights
- The noise precision
- The actual number of iterations to achieve the stopping criterion

## F   MODEL SPACE

We now provide details on the model space used to study the meta-learning problem formulated in our work. Recall that a model in the context of our problem is a time-series forecasting algorithm and the hyperparameters used.

**DeepAR Salinas et al. (2020):** DeepAR experiments are using the model implementation provided by GluonTS version 1.7. We did grid search on different values of number of cells and the number of RNN layers hyperparameters of DeepAR since the defaults provided in GluonTS would often lead to apparently suboptimal performance on many of the datasets. The training parameters for each dataset are described in Table 9. All other parameters are defaults of gluonts.model.deepar.DeepAREstimator.

**Deep Factors Wang et al. (2019):** Deep Factors experiments are using the model implementation provided by GluonTS version 1.7. We did grid search over the number of units per hidden layer for the global RNN model and the number of global factors hyperparameters of Deep Factors. The training parameters for each dataset are described in Table 9. All other parameters are defaults of gluonts.model.deep_factor.DeepFactorEstimator.

Table 8: Time-series meta-features for characterizing an arbitrary time-series dataset. We extracts a comprehensive number of meta-features. The extracted meta-features are five categories: simple, statistical, information-theoretic meta-features, spectral, and landmarker meta-features. To the best of our knowledge, we emphasize that our landmarker meta-features are novel and that some components of spectral meta-features have not been used in any related work.

| Name | Formula | Property | Variants | No. of Features |
|---|---|---|---|---|
| Window Length | $|w_i|$ | Speed/Scalability | N/A | 1 |
| Number of Time-series variables | $v_i$ | Type | N/A | 1 |
| Lag Autocorrelation | $\rho_n$ | Feature Interdependence | $\rho_0 - \rho_9$ | 10 |
| Lag Partial Autocorrelation | $\alpha_k$ | Feature Interdependence | acf_agg | 15 |
| Standard Deviation Range | $\sigma > r(max - min)$ | Dispersion | $r \in [0.05, 0.95]$ | 20 |
| Maximum | $max_X$ | Data Range | max_duplicates | 2 |
| Minimum | $min_X$ | Data Range | min_duplicates | 2 |
| Peaks | $peak_X$ | Data Range | peaks_supports, no_peaks | 6 |
| Reoccurence Statistics | $\overline{X}$ | Data Range | reocc_sum,reocc_count,reocc_ratio | 5 |
| Median | $\mu$ | Concentration | rms | 2 |
| Mean | $\bar{X}$ | Concentration | rms | 2 |
| Variance | $\sigma^2$ | Dispersion | std_dev | 2 |
| Covariance | $Cov$ | Dispersion | benford_corr | 8 |
| Quantiles | $q_{0.1} - q_{0.9}$ | Dispersion | diff_quantiles | 10 |
| Mass Quantiles | $q\%_{0.1} - q\%_{0.9}$ | Dispersion | diff_quantiles | 10 |
| Count below Mean | $\sum \not{}(X < \mu)$ | Statistics | min,max,$\sigma$,$\mu$ | 5 |
| Count Above Mean | $\sum \not{}(X > \mu)$ | Statistics | min,max,$\sigma$,$\mu$ | 5 |
| Number of Crossings | $\sum_{i=1}^n 1_{[X==val]}$ | Statistics | zero_cross, one_cross, minus_cross, root_hyp_test | 5 |
| Kurtosis | $\frac{\mu_4}{\sigma^4}$ | Feature normality | sample_kurt | 2 |
| Skewness | Skw | Feature normality | sample_skew | 2 |
| Symm_looking | $|\mu_X - median_X| < r * (max_X - min_X)$ | Symmetry | $r \in [0.05, 0.95]$ | 12 |
| Reversal_Asymmetry | $\mathbb{E}[L^2(X)^2 \cdot L(X) - L(X) \cdot X^2]$ | Reversal Asymmetry | lag_L | 12 |
| Absolute Sum of Change | $\sum_{i=1}^n |x_{i+1} - x_i|$ | Difference | mean_chg, mean_abs_chg, cid | 5 |
| Change Quantiles | Corridor Quantiles | Difference | $q_l \in [0, 1], q_h \in [q_l, 1]$ | 60 |
| Entropy | $D$ | Regularity | approx_entropy, sample_entropy | 21 |
| Linear Trend | Linear_reg_chunks | Trend | {"pvalue", "rvalue", "intercept", "slope", "stderr"} | 50 |
| Non-Linearity | $c_3$ | Non-linearity | $\ell \in \{1, 2, 3\}$, matrix_profile_feat | 9 |
| Friedrich Coefficients | $x' = h(x)$ | Dynamics | Langevin_Coeffs | 5 |
| Wavelet Transform Coefficients | Mexican hat wavelet | Time-Freq | "widths", "coeffs" | 60 |
| Fast Fourier Transform Coeeficients | FFT(X) | Frequency | {"real", "imag", 'agg_metrics'} | 201 |
| Polar Fast Fourier Transform Coeeficients | FFT(X) | Frequency | {"abs", "angle", 'agg_metrics'} | 201 |
| Absolute Energy | $E$ | Spectral | Energy_ratio_chunks, cross_pw_spect_dens | 15 |
| Fourier Entropy | $D_F$ | Spectral Regularity | binned_entropy | 1 |
| (1) Auto Regression | | | | |
| Regression Coefficients | See Appendix E | Landmarker | $\ell \in \{1, \cdots, 10\}$ | 10 |
| (2) Random Forest | | | | |
| Number of Leaves | See Appendix E | Landmarker | max, min, mean, std, skew, kurtosis | 6 |
| Tree depth | See Appendix E | Landmarker | max, min, mean, std, skew, and kurtosis | 6 |
| Mean of Base Tree Feature Importance | See Appendix E | Landmarker | max, min, mean, std, skew, and kurtosis | 6 |
| Max of Base Tree Feature Importance | See Appendix E | Landmarker | max, min, mean, std, skew, and kurtosis | 6 |
| Out-of-bag estimate score | See Appendix E | Landmarker | N/A | 1 |
| (3) Bayesian Ridge Regression | | | | |
| Mean of Distribution | See Appendix E | Landmarker | N/A | 1 |
| Log Marginal Likelihood | See Appendix E | Landmarker | N/A | 1 |
| Estimated Weights' Precision | See Appendix E | Landmarker | N/A | 1 |
| Estimated Noise Precision | See Appendix E | Landmarker | N/A | 1 |
| Stopping Number of Iterations | See Appendix E | Landmarker | N/A | 1 |
| | | | | 810 |

Table 9: Time-Series Forecasting Model Space. See hyperparameter definitions for various algorithms from GluonTS (Alexandrov et al., 2020) and statsmodels (Seabold & Perktold, 2010).

| Forecasting Algorithm | HyperParameter 1 | HyperParameter 2 | Data Representation | Total |
|---|---|---|---|---|
| DeepAR | num_cells = [10,20,30,40,50] | num_rnn_layers = [1,2,3,4,5] | {Exp_smoothing, Raw} | 50 |
| DeepFactor | num_hidden_global = [10,20,30,40,50] | num_global_factors = [1,5,10,15,20] | {Exp_smoothing, Raw} | 50 |
| Prophet | changepoint_prior_scale = [0.001, 0.01, 0.1, 0.2, 0.5] | seasonality_prior_scale = [0.01, 0.1, 1.0, 5.0, 10.0] | {Exp_smoothing, Raw} | 50 |
| Seasonal Naive | season_length = [1,5,7,10,30] | N/A | {Exp_smoothing, Raw} | 10 |
| Gaussian Process | cardinality = [2,4,6,8,10] | max_iter_jitter = [5,10,15,20,25] | {Exp_smoothing, Raw} | 50 |
| Vector Auto Regression | cov_type= {"HC0","HC1","HC2","HC3","nonrobust"} | trend = {'n', 'c', 't', 'ct' } | {Exp_smoothing, Raw} | 40 |
| Random Forest Regressor | n_estimators = [10,50,100,250,500,1000] | max_depth = [2,5,10,25,50,'None'] | {Exp_smoothing, Raw} | 72 |
| | | | | 322 |

**Prophet Taylor & Letham (2018):** Prophet experiments are using the model python implementation provided by Facebook (fbprophet) version 0.7.1. We did grid search over the change point prior scale and the seasonality prior scale hyperparameters of Prophet. The training parameters for each dataset are described in Table 9. All other parameters are defaults of fbprophet.Prophet.

**Seasonal Naive Montero-Manso et al. (2020):** Seasonal Naive experiments are using the model implementation provided by GluonTS version 1.7. We did grid search over the length of seasonality pattern, since it is different unknown for each, dataset hyperparameter of Seasonal Naive. The training parameters for each dataset are described in Table 9. All other parameters are defaults of gluonts.model.seasonal_naive.SeasonalNaivePredictor.

**Gaussian Process Yan et al. (2009):** Gaussian Process experiments are using the model implementation provided by GluonTS version 1.7. We did grid search over the cardinality of the time-series and the maximum number of iterations for jitter to iteratively make the matrix positive definite hyperparameter of Gaussian Proces. The training parameters for each dataset are described in Table 9. All other parameters are defaults of gluonts.model.gp_forecaster.GaussianProcessEstimator.

**Vector Auto Regression Lewis & Reinsel (1985):** Vector Auto Regression experiments are using the model implementation provided by statsmodels python library version 0.12.2. We did grid search over the loss covariance type and the trend hyperparameter of Vector Auto Regression. The training parameters for each dataset are described in Table 9. All other parameters are defaults of statsmodels.tsa.var_model.

**Random Forest Liaw et al. (2002):** Random Forest models' experiments are using the model implementation provided by sklearn python library version 0.24.2. We did grid search over the the number of estimators (trees) and the max_depth (i.e., the longest path between the root node and the leaf node in a tree) hyperparameter of Random Forest. The training parameters for each dataset are described in Table 9. All other parameters are defaults of sklearn.ensemble.RandomForestRegressor.

**Exponential Smoothing Data Representation Kalekar et al. (2004):** We use the exponential smoothing from the statsmodels python library version 0.12.2. We optimized the smoothing level for the exponential smoothing (which controls the weights given for the history samples). All other parameters are defaults of statsmodels.tsa.api.ExponentialSmoothing.

# G DATASET DESCRIPTION

## G.1 DATASET TYPES IN AUTOFORECAST

In our work, we consider two general types of time-series datasets depending on the number of the variables $v_i$ in the time-series dataset. Now, we detail the different dataset types.

**Univariate Time-series Datasets**: This kind of datasets is the traditional single time-series which usually consists of single variable that need to be predicted (this variable is likely numerical, but can also be categorical in rare cases). Formally, a time-series dataset $D_i$ is single-variate if $v_i = 1$.

**Multivariate Time-series Datasets**: In our work, we consider two main types (subcategories) of the Multivariate datasets that we define below. We define the **Multivariate Homogeneous dataset** as the multivariate dataset that consists of multiple time-series in which each time-series represents the same metric (*e.g.*, collection of $r$ time-series representing CPU usage of $r$ different machines). In this type, $v_i = r$ (the number of different time-series for that same measurement). Also, we define the **Multivariate Heterogeneous dataset** as multivariate time-series where each time-series column represents a different measurement (*e.g.*, wind speed, humidity, temperature). In this type, $v_i = |columns(D_i)|$ (the number of the different variables for the time-series dataset).

## G.2 DATASET TESTBEDS IN AUTOFORECAST

**Dataset Sources:** Our testbeds are built to simulate the testbed when meta-train comes from many different distributions. This diversity makes the meta-learning model learn from such diversity. Model selection on test data can thus benefit from the prior experience on the train set. For this purpose, we use benchmark datasets from Kaggle Kaggle (2021), Adobe real traces, and other open source repositories. In particular, the Adobe trace datasets records CPU and Memory usage for 50 different services running in Adobe production clusters collected for 15 days from May 1 to May 15 in 2021. Such traces are shared for the first time in our current work.

**Testbeds Summary:** In short, we have collected 308 univariate datasets from such different sources. The details of the datasets (i.e., the dataset name, and the number of points in the dataset) are shown in Table 10. We also collected 40 multivariate datasets, where each dataset can have different number of variables and different type. For instance, Adobe service CPU and memory 15 days utilization is a Homogeneous multivariate dataset that has $r = 100$. Most of the datasets are from different application domains (e.g., finance, IoT, energy, storage, etc.). The details of the datasets (i.e., the dataset name, the variate name, and the number of points for each variate the dataset) are detailed in Table 11. For robustness, for each testbed, we split its datasets into 5 folds for cross-validation. We build the train/test testbed by each time selecting four folds from the datasets for training and the remaining fifth fold for testing. We re-emphasize that we released the datstes for the community for future usage (link is in Section 1).

Table 10: Univariate Time-series dataset corpus description and details. The details of the datasets (i.e., the dataset name, and number of points in the dataset) are shown.

| Time-series Name | # pts | Time-series Name | # pts | Time-series Name | # pts | Time-series Name | # pts | Time-series Name | # pts |
|---|---|---|---|---|---|---|---|---|---|
| OLDMANT.1 | 1461 | ATT.1 | 97 | NONEMERG.1 | 319 | IV.1 | 44 | HOPEDALE.1 | 101 |
| WOLF.1 | 71 | FRASER.1 | 946 | EGDEMAN.2 | 100 | IBM.1 | 87 | LACSTJIN.1 | 1440 |
| HARBOR.1 | 157 | SERIESK.3 | 192 | PORKH.1 | 99 | construction | 319 | ESPANOLA.1 | 668 |
| RIOTIETE.1 | 372 | gdp_croatia | 24 | MINIMUM.1 | 848 | TURTLE.1 | 672 | OLDMAN.1 | 1470 |
| THAMES.1 | 71 | SERIESE.1 | 100 | sp500_price | 123 | SP500.1 | 99 | US.1 | 100 |
| Y.1 | 44 | BIGCONE.1 | 509 | LACSTJRA.1 | 1440 | FEEDH.1 | 95 | SERIESG.1 | 153 |
| FISHER.1 | 1470 | GLOBWARM.1 | 129 | TBILLS.1 | 102 | CCPI.1 | 102 | CAMPITO.1 | 5405 |
| JOE.3 | 1294 | CN.1 | 44 | SERIESJY.1 | 307 | bank | 581 | PRECIP.1 | 1096 |
| GNPR.1 | 85 | businv | 330 | MISINAB.1 | 672 | JUDITH.1 | 492 | TEMPER.1 | 1096 |
| RING.1 | 66 | LAKEVIEW.1 | 544 | DAILYSAP.1 | 3333 | GEODUCK.1 | 97 | HALSEY.1 | 108 |
| DAL.1 | 70 | SALESX.1 | 93 | AMAZON.2 | 55 | IP.1 | 111 | well_log | 675 |
| PCRGNP.1 | 62 | DS_Store | 137 | FEEDL.1 | 95 | DAILYIBM.1 | 3333 | SERIESC.1 | 228 |
| FRNCHB.1 | 45 | MAD.1 | 552 | UN.1 | 81 | JAMES.1 | 600 | CPI.1 | 288 |
| LYNDPIN.2 | 136 | OZONE.1 | 228 | GRANT.1 | 151 | SUNSPOTS.1 | 289 | KIEWA.1 | 72 |
| NEUMUNAS.1 | 132 | YD.1 | 44 | CHICKNYC.1 | 498 | NILEJJ.1 | 75 | robocalls | 52 |
| AMERICAN.1 | 660 | RHINE.1 | 150 | USM1.1 | 398 | PORKL.1 | 99 | ENGINES.1 | 188 |
| NIAGARA.1 | 1861 | YULE1.1 | 106 | SERIESA.1 | 200 | ASKEW4.1 | 660 | homeruns | 118 |
| SKUNK.1 | 71 | stocks_price | 560 | PRGNP.1 | 82 | OLDMANP.1 | 1507 | OOSTANAU.1 | 816 |
| DANUBE.1 | 120 | LACSTJSN.1 | 1440 | EMERGING.1 | 319 | CAFFEINE.1 | 178 | us_population | 816 |
| PLSUPER.1 | 104 | CIG.3 | 138 | SIMAR4.1 | 818 | SUNSPTMO.1 | 2820 | ENGLISH.1 | 660 |
| SERIESF.1 | 70 | BOISE.1 | 588 | FREEDMAN.1 | 58 | ASKEW3.1 | 708 | global_co2 | 104 |
| WHEAT.1 | 370 | EGGS.1 | 319 | VATNSD.1 | 1098 | SERIESJX.1 | 312 | QBIRTHS.1 | 5117 |
| MARTEN.1 | 71 | FOOD.1 | 178 | CONSUM.1 | 147 | FEED.1 | 95 | EXSHAW.1 | 506 |
| WHITEMTN.1 | 1164 | centralia | 15 | TSEOIL.1 | 361 | BRYCE.1 | 625 | MADISON.1 | 456 |
| apple | 622 | GOLDH.1 | 97 | MUSKRAT.1 | 71 | BAYDU.1 | 358 | jfk_passengers | 468 |
| SFSKYKOM.1 | 456 | SERIESD.1 | 312 | LOGISTIC.1 | 200 | VELMON.1 | 86 | PEAS.1 | 768 |
| OTTER_L.1 | 71 | METALS.1 | 178 | BIRTHS.1 | 59 | CURRENT.1 | 468 | ASKEW5.1 | 108 |
| ASKEW13.1 | 372 | ELBE.1 | 300 | FRNCHA.1 | 70 | PLHURON.1 | 104 | TOTAL.1 | 319 |
| unemployment_nl | 214 | BEARDS.1 | 67 | RIOGRAND.1 | 576 | occupancy | 509 | COLUM.1 | 444 |
| EMP.1 | 81 | RGNP.1 | 85 | rail_lines | 37 | PIPER.1 | 348 | GNPN.1 | 85 |
| SPIRITS.3 | 254 | NIGERIA.1 | 123 | MBOULDER.1 | 588 | debt_ireland | 21 | IPI.1 | 85 |
| CIGB.2 | 128 | GOLDL.1 | 97 | CMINEF.1 | 96 | NILE2.1 | 100 | ozone | 54 |
| ASKEW7.1 | 600 | GLOBTP.1 | 136 | gdp_iran | 58 | TIOGA.1 | 661 | SERIESB.1 | 385 |
| G.1 | 46 | MCKEN.1 | 55 | children_per_woman | 301 | YULE2.1 | 107 | ISH66.1 | 163 |
| GOTA.1 | 150 | iot_temp | 8402 | PPHIL.1 | 1572 | WOODS.1 | 629 | DJWEEK.1 | 186 |
| SAUGEEN.1 | 1403 | USH.1 | 100 | nile | 100 | co2_canada | 215 | NAVAJO.1 | 700 |
| TRADE.1 | 178 | GRUEN.1 | 53 | PACK.2 | 344 | bee_waggle_6 | 609 | ARCTIC.1 | 66 |
| whin_temp | 6074 | shanghai_license | 205 | WG.1 | 71 | NYSE.1 | 87 | AZUSA.1 | 180 |
| SERIESL.2 | 549 | SOY.1 | 99 | FORTALEZ.1 | 150 | NMAGNET.1 | 732 | AROSA.1 | 480 |
| FURNAS.DAT | 576 | LAKEMICH.1 | 115 | MCKENZIE.1 | 600 | ROCKY.1 | 122 | SNOW.1 | 54 |
| GUELPH.1 | 72 | gdp_argentina | 59 | SOYL.1 | 99 | SAUGEENP.1 | 1412 | PORK.1 | 99 |
| uk_coal_employ | 105 | ASKEW9.1 | 588 | ASKEW14.1 | 588 | TRINITY.1 | 588 | OGDEN.1 | 97 |
| TPMON.1 | 2976 | I.1 | 44 | DEATHS.1 | 319 | SSASK.1 | 780 | SOYH.1 | 99 |
| SAUGEENT.1 | 1412 | SKIRTS.1 | 69 | M.1 | 85 | OKAK.1 | 109 | EAGLECOL.1 | 858 |
| ASKEW10.1 | 600 | WOLVEREN.1 | 71 | SCHOLES.1 | 114 | KINGS.1 | 49 | STJOHNS.1 | 600 |
| SERIESJ.2 | 616 | NILEMON.1 | 910 | BOXHU1.1 | 48 | YEAR.1 | 208 | BOXHUN.1 | 217 |
| CORN.2 | 76 | GNP.1 | 62 | SUMMER.1 | 208 | BLUME.1 | 64 | NARAMATA.1 | 515 |
| HURON.1 | 157 | USM2.1 | 398 | CRYER.1 | 43 | usd_isk | 247 | MSTOUIS.1 | 96 |
| MEASLNYC.1 | 534 | seatbelts | 192 | ASKEW12.1 | 456 | NILE.1 | 75 | ALIGN.1 | 55 |
| USL.1 | 100 | ELECUS.1 | 51 | SERIESB2.1 | 271 | brent_spot | 500 | measles | 991 |
| RWG.1 | 71 | SNAKE.1 | 669 | REDDEER.1 | 396 | MUMPS.1 | 534 | CANFIRE.1 | 71 |
| CMINER.1 | 528 | CLEARWAT.1 | 600 | bitcoin | 774 | NYWATER.1 | 71 | ASKEW15.1 | 432 |
| ASKEW.1 | 264 | NECHES.1 | 564 | RACOON.1 | 71 | ASKEW8.1 | 456 | CD.1 | 44 |
| RAPPAHAN.1 | 600 | AARIVINT.1 | 213 | gdp_japan | 58 | WBDELAWA.1 | 540 | TPYR.1 | 248 |
| VEL.1 | 102 | TRANEQ.1 | 178 | NINEMILE.1 | 771 | HEBRON.1 | 109 | NAMAKAN.1 | 648 |
| DELL.1 | 655 | PLMICH.1 | 104 | SERIESM.2 | 300 | MINK.1 | 71 | run_log | 376 |
| SUNSPT.1 | 261 | CO2.1 | 192 | PAPER.2 | 320 | FEATHER.1 | 708 | FLOW.1 | 468 |
| FISHERT.1 | 1471 | PREC.1 | 136 | JOKULSA.1 | 1096 | HBCO.1 | 66 | WINTER.1 | 208 |
| NAIN.1 | 109 | CMINET.1 | 528 | WATERQ.1 | 147 | MEASLBAL.1 | 402 | BND.1 | 71 |
| LYNX.1 | 154 | GOLD.1 | 97 | lga_passengers | 468 | PIGEON.1 | 636 | USM3.1 | 398 |
| FISHERP.1 | 1471 | IRONSU.3 | 143 | BWATER.1 | 79 | HANKOU.1 | 1368 | DVI.1 | 470 |
| RICHELU.1 | 468 | DJ.1 | 157 | U.1 | 85 | | | | |

# H   DISCUSSION ON INFERENCE TIME COMPLEXITY

Recall that the number of meta-features is $d$, the number of models is $m$. The time complexity for the inference part of the general meta-learner $\Phi$ is $\mathcal{O}(d)$. On the other hand, the time complexity of the time-series LSTM meta-learner $\Theta$ is given by $\mathcal{O}(|X_t|)$, where $|X_t|$ is the length of the input sequence. Thus, AUTOFORECAST's inference time is given by $\max\{\mathcal{O}(d), \mathcal{O}(|X_t|)\}$. We emphasize that the inference time of the naïve method is much larger since it is given by $\mathcal{O}(\sum_{i=1}^{m} I_i)$, where $I_i$ is the inference time of forecasting algorithm $a_i$. The aggregate statistics for reduction in inference time across both testbeds is shown in Appendix J.5. Moreover, Figure 7 shows that AUTOFORECAST yields significant reduction in inference time (68X reduction) over the naïve approach on the Adobe trace CPU and Memory 15 days usage dataset.

Table 11: Multivariate Time-series dataset corpus description and details (i.e., the dataset name, the variate name, and the number of points for each variate of the dataset).

| Dataset Name | Variable Name | # pts | Dataset Name | Variable Name | # pts | Dataset Name | Variable Name | # pts |
|---|---|---|---|---|---|---|---|---|
| Processed_S&P | Oil | 1984 | Processed_S&P | Nikkei-F | 1984 | Processed_S&P | ROC_5 | 1984 |
| Processed_S&P | NASDAQ-F | 1984 | Processed_S&P | mom1 | 1984 | Processed_S&P | mom3 | 1984 |
| Processed_S&P | mom2 | 1984 | Processed_S&P | MSFT | 1984 | Processed_S&P | NZD | 1984 |
| Processed_S&P | NYSE | 1984 | Processed_S&P | ROC_10 | 1984 | | | |
| ozone_onehr | T9 | 2536 | ozone_onehr | T8 | 2536 | ozone_onehr | T1 | 2536 |
| ozone_onehr | T3 | 2536 | ozone_onehr | T2 | 2536 | ozone_onehr | T6 | 2536 |
| ozone_onehr | T7 | 2536 | ozone_onehr | T5 | 2536 | ozone_onehr | T4 | 2536 |
| ozone_onehr | T12 | 2536 | ozone_onehr | T11 | 2536 | ozone_onehr | T10 | 2536 |
| energydata_complete | RH_1 | 19735 | energydata_complete | Press_mm_hg | 19735 | energydata_complete | lights | 19735 |
| energydata_complete | Appliances | 19735 | energydata_complete | RH_2 | 19735 | energydata_complete | RH_3 | 19735 |
| energydata_complete | RH_1 | 19735 | energydata_complete | Visibility | 19735 | energydata_complete | Windspeed | 19735 |
| Sales_Transactions | p19 | 52 | Sales_Transactions | p18 | 52 | Sales_Transactions | p20 | 52 |
| Sales_Transactions | p1 | 52 | Sales_Transactions | p2 | 52 | Sales_Transactions | p3 | 52 |
| Sales_Transactions | p7 | 52 | Sales_Transactions | p6 | 52 | Sales_Transactions | p4 | 52 |
| Sales_Transactions | p5 | 52 | Sales_Transactions | p8 | 52 | Sales_Transactions | p9 | 52 |
| Sales_Transactions | p10 | 52 | Sales_Transactions | p11 | 52 | Sales_Transactions | p13 | 52 |
| Sales_Transactions | p12 | 52 | Sales_Transactions | p16 | 52 | Sales_Transactions | p17 | 52 |
| Sales_Transactions | p15 | 52 | Sales_Transactions | p14 | 52 | | | |
| AdobeAveCPU_96x3270 | S40 | 96 | AdobeAveCPU_96x3270 | S36 | 96 | AdobeAveCPU_96x3270 | S37 | 96 |
| AdobeAveCPU_96x3270 | S31 | 96 | AdobeAveCPU_96x3270 | S33 | 96 | AdobeAveCPU_96x3270 | S32 | 96 |
| AdobeAveCPU_96x3270 | S4 | 96 | AdobeAveCPU_96x3270 | S6 | 96 | AdobeAveCPU_96x3270 | S38 | 96 |
| AdobeAveCPU_96x3270 | S1 | 96 | AdobeAveCPU_96x3270 | S20 | 96 | | | |
| fast-storage-20 | Memory capacity provisioned | 8615 | fast-storage-20 | Network received throughput | 8615 | fast-storage-20 | Network transmitted throughput | 8615 |
| fast-storage-20 | Disk write throughput | 8615 | fast-storage-20 | CPU capacity provisioned | 8615 | fast-storage-20 | CPU cores | 8615 |
| fast-storage-20 | Timestamp | 8615 | fast-storage-20 | CPU usage | 8615 | fast-storage-20 | Disk read throughput | 8615 |
| fast-storage-20 | Memory usage | 8615 | | | | | | |
| knoy_mpu_3_300 | Y | 599 | knoy_mpu_3_300 | X | 599 | knoy_mpu_1_400 | Y | 1230 |
| knoy_mpu_1_400 | X | 1230 | knoy_mpu_1_340 | Y | 1709 | knoy_mpu_1_340 | X | 1709 |
| Scanline | scanline_42049 | 481 | Scanline | scanline_126007 | 481 | | | |
| iowa-electricity | net_generation | 51 | iowa-electricity | local_generation | 51 | | | |
| Adobe_CPU_Mem_15d | stageva6–STGusedmem | 2016 | Adobe_CPU_Mem_15d | stageirl1–QA2usedcpu | 2016 | Adobe_CPU_Mem_15d | stageirl1–QA2usedmem | 2016 |
| Adobe_CPU_Mem_15d | stageva6–STG1usedmem | 2016 | Adobe_CPU_Mem_15d | stageirl1–QAusedcpu | 2016 | Adobe_CPU_Mem_15d | stageva6–STG1usedcpu | 2016 |
| Adobe_CPU_Mem_15d | stageirl1–QA2usedmem | 2016 | Adobe_CPU_Mem_15d | stageirl1–QAusedmem | 2016 | Adobe_CPU_Mem_15d | stageva6–STG1usedcpu | 2016 |
| Adobe_CPU_Mem_15d | prodjpn3–PRODusedmem | 2014 | Adobe_CPU_Mem_15d | stageirl1–Stageusedmem | 2016 | Adobe_CPU_Mem_15d | stageva6–QAusedmem | 2016 |
| Adobe_CPU_Mem_15d | stageirl1–QAusedcpu | 2016 | Adobe_CPU_Mem_15d | stageirl1–STG10usedcpu | 2016 | Adobe_CPU_Mem_15d | stageirl1–STG10usedmem | 2016 |
| Adobe_CPU_Mem_15d | stageirl1–QAusedmem | 2016 | Adobe_CPU_Mem_15d | stageva6–QAusedcpu | 2016 | Adobe_CPU_Mem_15d | prodjpn3–PRODusedmem | 2014 |
| Adobe_CPU_Mem_15d | stageirl1–Stageusedcpu | 2016 | Adobe_CPU_Mem_15d | prodjpn3–PROD1usedmem | 2014 | Adobe_CPU_Mem_15d | prodjpn3–Productionusedmem | 2014 |
| Adobe_CPU_Mem_15d | stageva6–STG1usedmem | 2016 | Adobe_CPU_Mem_15d | prodirl1–PRODusedcpu | 2016 | Adobe_CPU_Mem_15d | prodjpn3–PROD1usedcpu | 2014 |
| Adobe_CPU_Mem_15d | prodva6–PROD1usedmem | 2016 | Adobe_CPU_Mem_15d | prodjpn3–Productionusedcpu | 2014 | Adobe_CPU_Mem_15d | prodjpn3–PROD1usedcpu | 2014 |
| Adobe_CPU_Mem_15d | prodirl1–PRODusedmem | 2016 | Adobe_CPU_Mem_15d | stageva6–STG1usedcpu | 2016 | Adobe_CPU_Mem_15d | stageva6–QAusedmem | 2016 |
| Adobe_CPU_Mem_15d | prodirl1–PROD1usedmem | 2016 | Adobe_CPU_Mem_15d | prodirl1–PRODusedmem | 2016 | Adobe_CPU_Mem_15d | stageirl1–QAusedmem | 2016 |
| Adobe_CPU_Mem_15d | stageirl1–QAusedcpu | 2016 | Adobe_CPU_Mem_15d | stageva6–QAusedcpu | 2016 | Adobe_CPU_Mem_15d | prodirl1–PROD1usedmem | 2016 |
| Adobe_CPU_Mem_15d | prodirl1–PRODusedcpu | 2016 | Adobe_CPU_Mem_15d | prodjpn3–PRODusedcpu | 2014 | Adobe_CPU_Mem_15d | stageirl1–QA2usedcpu | 2016 |
| Adobe_CPU_Mem_15d | stageva6–QAusedmem | 2016 | Adobe_CPU_Mem_15d | prodjpn3–PROD10usedcpu | 2014 | Adobe_CPU_Mem_15d | prodjpn3–PROD10usedmem | 2014 |
| Adobe_CPU_Mem_15d | stageva6–QAusedmem | 2016 | Adobe_CPU_Mem_15d | stageirl1–QA2usedmem | 2016 | Adobe_CPU_Mem_15d | prodjpn3–PRODusedmem | 2014 |
| Adobe_CPU_Mem_15d | stageirl1–QA2usedcpu | 2016 | Adobe_CPU_Mem_15d | stageirl1–QA2usedmem | 2016 | Adobe_CPU_Mem_15d | stageva6–QAusedmem | 2016 |
| Adobe_CPU_Mem_15d | stageirl1–QA2usedmem | 2016 | Adobe_CPU_Mem_15d | stageirl1–QAusedmem | 2016 | Adobe_CPU_Mem_15d | stageva6–QA1usedmem | 2016 |
| Adobe_CPU_Mem_15d | stageva6–QAusedmem | 2016 | Adobe_CPU_Mem_15d | prodva6–PROD1usedcpu | 2016 | Adobe_CPU_Mem_15d | prodirl1–PROD1usedcpu | 2016 |
| Adobe_CPU_Mem_15d | stageva6–QA1usedcpu | 2016 | Adobe_CPU_Mem_15d | stageva6–QAusedcpu | 2016 | Adobe_CPU_Mem_15d | stageva6–QAusedcpu | 2016 |
| Adobe_CPU_Mem_15d | stageva6–QAusedmem | 2016 | Adobe_CPU_Mem_15d | prodva6–PROD1usedmem | 2016 | Adobe_CPU_Mem_15d | prodirl1–PROD10usedmem | 2016 |
| Adobe_CPU_Mem_15d | stageirl1–QAusedmem | 2016 | Adobe_CPU_Mem_15d | stageirl1–QAusedmem | 2016 | Adobe_CPU_Mem_15d | stageirl1–QAusedmem | 2016 |
| Adobe_CPU_Mem_15d | stageirl1–QAusedmem | 2016 | Adobe_CPU_Mem_15d | stageirl1–QA2usedmem | 2016 | Adobe_CPU_Mem_15d | stageirl1–QA2usedcpu | 2016 |
| Adobe_CPU_Mem_15d | stageirl1–QA2usedmem | 2016 | Adobe_CPU_Mem_15d | stageirl1–QA2usedmem | 2016 | Adobe_CPU_Mem_15d | stageirl1–STGusedmem | 2016 |
| Adobe_CPU_Mem_15d | stageva6–QAusedcpu | 2016 | Adobe_CPU_Mem_15d | stageirl1–STGusedcpu | 2016 | Adobe_CPU_Mem_15d | stageva6–QAusedmem | 2016 |
| Adobe_CPU_Mem_15d | prodva6–PRODusedmem | 2016 | Adobe_CPU_Mem_15d | stageirl1–STGusedmem | 2016 | Adobe_CPU_Mem_15d | stageirl1–QA10usedmem | 2016 |
| Adobe_CPU_Mem_15d | stageirl1–STGusedcpu | 2016 | Adobe_CPU_Mem_15d | stageirl1–QA10usedcpu | 2016 | Adobe_CPU_Mem_15d | prodva6–PRODusedcpu | 2016 |
| Adobe_CPU_Mem_15d | prodirl1–Productionusedmem | 2016 | Adobe_CPU_Mem_15d | stageirl1–QA10usedmem | 2016 | Adobe_CPU_Mem_15d | stageva6–STGusedmem | 2016 |
| Adobe_CPU_Mem_15d | stageirl1–STGusedmem | 2016 | Adobe_CPU_Mem_15d | prodva6–Productionusedmem | 2016 | Adobe_CPU_Mem_15d | stageva6–STGusedcpu | 2016 |
| Adobe_CPU_Mem_15d | prodva6–Productionusedcpu | 2016 | Adobe_CPU_Mem_15d | stageirl1–STGusedcpu | 2016 | Adobe_CPU_Mem_15d | prodirl1–Productionusedcpu | 2016 |
| Adobe_CPU_Mem_15d | stageirl1–QA10usedcpu | 2016 | Adobe_CPU_Mem_15d | stageirl1–STG1usedcpu | 2016 | Adobe_CPU_Mem_15d | stageirl1–QAusedcpu | 2016 |
| Adobe_CPU_Mem_15d | prodva6–PRODusedmem | 2016 | Adobe_CPU_Mem_15d | stageirl1–STGusedmem | 2016 | Adobe_CPU_Mem_15d | stageirl1–STGusedmem | 2016 |
| Adobe_CPU_Mem_15d | prodva6–PRODusedmem | 2016 | Adobe_CPU_Mem_15d | stageirl1–STG1usedmem | 2016 | Adobe_CPU_Mem_15d | stageirl1–QAusedmem | 2016 |
| Adobe_CPU_Mem_15d | stageva6–STGusedmem | 2016 | | | | | | |
| ozone_eighthr | HT70 | 2534 | ozone_eighthr | SLP | 2534 | ozone_eighthr | Precp | 2534 |
| ozone_eighthr | RH70 | 2534 | ozone_eighthr | RH85 | 2534 | ozone_eighthr | RH50 | 2534 |
| ozone_eighthr | KI | 2534 | ozone_eighthr | SLP | 2534 | ozone_eighthr | HT85 | 2534 |
| ozone_eighthr | HT50 | 2534 | | | | | | |
| quality_control | 4 | 500 | quality_control | 5 | 325 | quality_control | 2 | 283 |
| quality_control | 3 | 366 | quality_control | 1 | 313 | | | |
| knoy_mpu_3_400 | X | 720 | knoy_mpu_3_400 | Y | 720 | knoy_mpu_2_400 | X | 1546 |
| knoy_mpu_2_400 | Y | 1546 | knoy_mpu_1_500 | Y | 2871 | knoy_mpu_1_500 | X | 2871 |
| knoy_mpu_3_100 | Y | 824 | knoy_mpu_3_100 | X | 824 | knoy_mpu_1_360 | Y | 1252 |
| knoy_mpu_1_360 | X | 1252 | knoy_mpu_2_100 | Y | 757 | knoy_mpu_2_100 | X | 757 |
| knoy_mpu_2_500 | X | 1605 | knoy_mpu_2_500 | Y | 1605 | knoy_mpu_1_100 | X | 1215 |
| knoy_mpu_1_100 | Y | 1215 | knoy_mpu_3_380 | X | 574 | knoy_mpu_3_380 | Y | 574 |
| knoy_mpu_2_320 | Y | 887 | knoy_mpu_2_320 | X | 887 | knoy_mpu_2_380 | X | 848 |
| knoy_mpu_2_380 | X | 848 | knoy_mpu_1_380 | Y | 1499 | knoy_mpu_1_380 | X | 1499 |
| Processed_NASD | DTB4WK | 1984 | Processed_NASD | EMA_50 | 1984 | Processed_NASD | DTB3 | 1984 |
| Processed_NASD | DTB6 | 1984 | Processed_NASD | EMA_20 | 1984 | Processed_NASD | FCHI | 1984 |
| Processed_NASD | FTSE-F | 1984 | Processed_NASD | EMA_10 | 1984 | Processed_NASD | EMA_200 | 1984 |
| Processed_NASD | EUR | 1984 | | | | | | |
| Occupancy_CO2 | Occupancy | 8143 | Occupancy_CO2 | Temperature | 8143 | Occupancy_CO2 | CO2 | 8143 |
| Occupancy_CO2 | Humidity | 8143 | Occupancy_CO2 | Light | 8143 | | | |
| us-employment | financial_activities | 120 | us-employment | nonfarm_change | 120 | us-employment | construction | 120 |
| us-employment | mining_and_logging | 120 | us-employment | information | 120 | us-employment | professional_and_business_services | 120 |
| us-employment | durable_goods | 120 | | | | | | |
| Occ_train | txt_Light | 8143 | Occ_train | txt_CO2 | 8143 | Occ_train | txt_Humidity | 8143 |
| Occ_train | txt_HumidityRatio | 8143 | Occ_train | txt_Temperature | 8143 | Occ_train | txt_Occupancy | 8143 |
| Processed_DJI | DAX-F | 1984 | Processed_DJI | DE6 | 1984 | Processed_DJI | DGS10 | 1984 |
| Processed_DJI | DE5 | 1984 | Processed_DJI | DE4 | 1984 | Processed_DJI | DE1 | 1984 |
| Processed_DJI | DE2 | 1984 | Processed_DJI | DAAA | 1984 | Processed_DJI | DGS5 | 1984 |
| Processed_DJI | DBAA | 1984 | | | | | | |
| Processed_RUSS | Brent | 1984 | Processed_RUSS | AUD | 1984 | Processed_RUSS | AAPL | 1984 |
| Processed_RUSS | CNY | 1984 | Processed_RUSS | Close | 1984 | Processed_RUSS | CAD | 1984 |
| Processed_RUSS | copper-F | 1984 | Processed_RUSS | CHF | 1984 | Processed_RUSS | AMZN | 1984 |
| Processed_RUSS | CAC-F | 1984 | | | | | | |
| Processed_NYSE | IXIC | 1984 | Processed_NYSE | JNJ | 1984 | Processed_NYSE | gold-F | 1984 |
| Processed_NYSE | Gold | 1984 | Processed_NYSE | JPM | 1984 | Processed_NYSE | GBP | 1984 |
| Processed_NYSE | GE | 1984 | Processed_NYSE | GDAXI | 1984 | Processed_NYSE | HSI-F | 1984 |
| Processed_NYSE | HSI | 1984 | | | | | | |
| knoy_mpu_3_600 | X | 876 | knoy_mpu_3_600 | Y | 876 | knoy_mpu_2_600 | X | 342 |
| knoy_mpu_2_600 | Y | 344 | knoy_mpu_2_200 | Y | 765 | knoy_mpu_2_200 | X | 765 |
| knoy_mpu_1_600 | Y | 2321 | knoy_mpu_1_600 | X | 2321 | knoy_mpu_3_200 | Y | 845 |
| knoy_mpu_3_200 | X | 845 | | | | | | |
| co2-concentration | CO2 | 741 | co2-concentration | adjusted CO2 | 741 | | | |
| fast-storage-1 | Disk write throughput | 8634 | fast-storage-1 | Network received throughput | 8634 | fast-storage-1 | Memory capacity provisioned | 8634 |
| fast-storage-1 | Memory usage | 8634 | fast-storage-1 | CPU capacity provisioned | 8634 | fast-storage-1 | CPU cores | 8634 |
| fast-storage-1 | Disk read throughput | 8634 | fast-storage-1 | Network transmitted throughput | 8634 | fast-storage-1 | CPU usage | 8634 |

# I    BASELINES DETAILS

Meta-learning has recently received a significant attention for automating ML pipelines for a variety of different problems outside the domain of time-series forecasting including supervised learning Feurer et al. (2015); Wistuba et al. (2018), classification and regression Finn et al. (2017); Rusu et al. (2019), unsupervised learning Abdulrahman et al. (2018), outlier detection Zhao et al. (2020) and other applications Mittal et al. (2020); Vinyals et al. (2016).

In particular, meta-learning has been leveraged for such automation by designing models for new tasks based on prior experience Vanschoren (2018); Yao et al. (2018); Raghu et al. (2020).

## I.1 BASELINE CATEGORIES

We now detail each such baseline. Most of the methods proposed in these works can be categorized into the following three main categories.[4]

**No model selection**: This category always employs either the same single model or the ensemble of all the models:

- **Random Forest (RF)** Liaw et al. (2002): is a SOTA tree ensemble that combines the predictions made by many decision trees into a single model. In prediction, the random forest regression model takes the average of all the individual decision tree estimates.

**Simple meta-learners**: Meta-learners in this category pick the generally well-performing forecasting model, globally or locally:

- **Global Best (GB)**: is the simplest meta-learner that selects the forecasting model with the largest average performance across all train datasets (across all time windows), without using any meta-features.

- **ISAC** Kadioglu et al. (2010): clusters the meta-train datasets based on meta-features. Given a new test time-series dataset, it identifies its closest cluster and selects the best model with largest average performance on the cluster's datasets.

- **ARGOSMART (AS)** Nikolić et al. (2013): finds the closest meta-train time-series dataset (1NN) to a given test time-series dataset, based on meta-feature similarity, and selects the model with the best performance on the 1NN dataset.

**Optimization-based meta-learners**: Meta-learners in this category learn meta-feature by task similarities toward optimizing performance estimates:

- **Multi-layer Perceptron (MLP)**: Given the meta-train datasets and selected time window, the MLP regressor directly maps the meta-features onto model performances by regression. However, such baseline does not learn temporal dependence within datasets.

- **AUTOFORECAST-TSL**: is a variant in which the meta-learner $\mathcal{L}$ consists only of the time-series learner $\Theta$.

## I.2 USAGE OF META-LEARNING IN AUTOFORECAST

We emphasize that we use the term "meta-learning" in the context of traditional principle of meta-learning which is building upon prior experience on a set of historical tasks to "do better" on a new task. We build the experience across different datasets using our "general meta-learner" and build experience on the sequential dependence among the same dataset using the "LSTM time-series meta-learner". We also capture task similarity between a new input task (dataset) and historical datasets using the "meta-features".

We also emphasize that our proposed method is faster compared to gradient-descent-based bi-level meta-learners, e.g., on our univariate tesbed N-Beats has significantly slower training (average = 3600 seconds) and inference time (average = 101 seconds) compared to AUTOFORECAST (average = 670 seconds for training and 1.13 seconds for inference). We emphasize that enhancing gradient-descent-based bi-level meta-learners for our new problem is an avenue for future work.

---

[4]As will be shown in Table 14, we emphasize that we did a model selection on seven popular time series forecasting models as well (including the recent works DeepAR Salinas et al. (2020), DeepFactors Wang et al. (2019), and Prophet Taylor & Letham (2018)) as baselines where we chose the model with the best average performance across all windows for each dataset from all model variants.

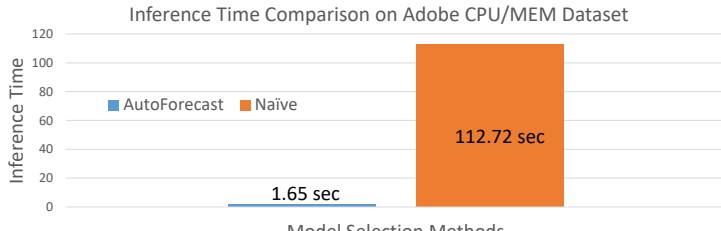

Figure 7: A comparison of inference time between AUTOFORECAST and Naïve approach (running inference on all 322 models and selecting the best candidate) on Adobe CPU/Mem trace 15 days. AUTOFORECAST yields significant reduction in inference time (68X reduction).

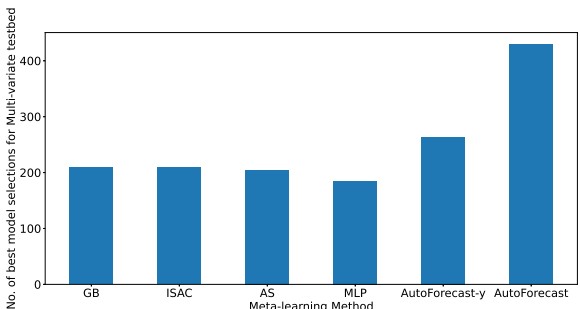

Figure 8: The number of the best model selections by each meta-learning approach for multivariate testbed. AUTOFORECAST has 2X gain in selecting best model compared to other baselines.

## J EXTENDED RESULTS

### J.1 FULL EVALUATION ON DATASETS PER TESTBED

**Univariate Testbed:** We now present the full evaluation of univariate testbed. Such evaluation is shown in Table 12 (we show one fold in that table in the interest of space). We note similar performances for the rest of the folds (as reflected in the average MSE and average rank for all datasets shown in Table 3-4). AUTOFORECAST outperforms the meta-learner baselines for most datasets. Moreover, AUTOFORECAST has the lowest average MSE, and the lowest average rank.

**Multivariate Testbed:** We here present the full evaluation of multivariate testbed. Such evaluation is shown in Table 13. It is clear that AUTOFORECAST ouperforms the meta-learner baselines for most datasets (best for 28 out of the 40 datasets). Again, we note that AUTOFORECAST has the lowest average MSE, and the lowest average rank. In particular, AUTOFORECAST (MSE = 0.12) gives a gain of 50% over the best baseline (MSE = 0.26) on the Adobe_CPU_Mem_15d dataset.

**Meta-learners perform better than methods without model selection:** Table 14 shows that meta-learners outperforms almost all models with no model selection. In particular, three meta-learners (Global Best, ISAC, AUTOFORECAST) significantly outperform the baseline time-series forecasting models. For instance, Global Best respectively has 79.42%, 56.71%, 67.28%, 64.32%, 54.19%,95.73%, and 88.21% lower MSE over Seasonal Naive, DeepAR, Deep Factors, Random Forest, Prophet, Gaussian Process, and VAR. These results signify the benefits of model selection (specifically using AUTOFORECAST).

**Prophet and DeepAR have the best performance across the baseline forecasting algorithms:** Table 14 shows that Prophet model has the best average MSE across the baseline forecasting algorithms. Notably, DeepAR has the second best average MSE across them.

**Gain in Selecting better models using** AUTOFORECAST**:** Figure 8 shows that across the pool of the multivariate testbed (across all time windows), AUTOFORECAST has superiority in selecting better models compared to the baseline meta-learners (2X gain in selecting better forecasting models).

Table 12: Method evaluation in Univariate testbed (one fold is shown in the interest of space). The most performing (lowest MSE) method is highlighted in **bold**. The rank is provided in parenthesis (lower ranks denote better performance). AUTOFORECAST achieves the best average MSE and average rank among all meta-learners.

| Dataset | Global Best | AS | ISAC | MLP | AUTOFORECAST-TSL | AUTOFORECAST |
|---|---|---|---|---|---|---|
| **FEATHER.1** | 0.003429 (4) | 0.000916 (3) | 0.036634 (6) | 0.013679 (5) | **0.000716 (1)** | **0.000716 (1)** |
| **FURNAS.DAT** | 0.017221 (4) | 0.081947 (5) | 0.085216 (6) | 0.004338 (3) | **0.000177 (1)** | **0.000177 (1)** |
| **EGGS.1** | 0.0005 (3) | 0.000732 (4) | 0.006797 (5) | 0.02572 (6) | 0.000392 (2) | **0.000096 (1)** |
| **NYSE.1** | 0.041847 (4) | **0.011433 (1)** | 0.035967 (2) | 0.109728 (6) | 0.083661 (5) | 0.036996 (3) |
| **apple** | 0.000071 (3) | 0.000727 (5) | 0.000256 (4) | 0.000842 (6) | **0.000038 (1)** | **0.000038 (1)** |
| **STJOHNS.1** | 0.015178 (4) | 0.152463 (5) | **0.000281 (1)** | 0.51043 (6) | 0.010065 (3) | 0.009741 (2) |
| **SERIESM.2** | 0.00011 (3) | **0.000031 (1)** | 0.000081 (2) | 0.000254 (6) | 0.000197 (5) | 0.000184 (4) |
| **ALIGN.1** | 0.116045 (5) | **0.020837 (1)** | 0.158484 (6) | 0.10843 (4) | 0.07156 (3) | 0.031977 (2) |
| **LYNX.1** | 0.029034 (3) | 0.036677 (4) | 0.204602 (5) | 0.076905 (6) | 0.021614 (2) | **0.012232 (1)** |
| **ARCTIC.1** | 0.002078 (4) | 0.013534 (5) | 0.001891 (3) | 0.029793 (6) | 0.001175 (2) | **0.0006 (1)** |
| **RHINE.1** | **0.000992 (1)** | 0.030022 (6) | 0.026165 (5) | 0.011251 (3) | 0.013101 (4) | 0.008896 (2) |
| **ASKEW15.1** | 0.002235 (5) | 0.00036 (2) | 0.001733 (4) | 0.084906 (6) | 0.000881 (3) | **0.000182 (1)** |
| **MEASLBAL.1** | 0.000007 (2) | 0.012992 (5) | 0.000068 (3) | 0.073299 (6) | 0.00038 (4) | **0.000002 (1)** |
| **REDDEER.1** | 0.091479 (5) | 0.031401 (2) | 0.134347 (6) | 0.910554 (5) | **0.044033 (1)** | **0.044033 (1)** |
| **rail_lines** | **0.000043 (1)** | 0.105375 (5) | 0.001864 (4) | 0.1923 (6) | 0.000289 (2) | 0.000289 (2) |
| **data_temp_dev** | 0.00011 (2) | 0.000145 (3) | **0.000096 (1)** | 0.03142 (6) | 0.000335 (4) | 0.000335 (4) |
| **DELL.1** | 0.005872 (3) | 0.008785 (4) | 0.013155 (5) | 0.196153 (6) | **0.00209 (1)** | **0.00209 (1)** |
| **TEMPER.1** | 0.015079 (4) | 0.011044 (3) | 0.015079 (5) | 0.489135 (6) | **0.006356 (1)** | **0.006356 (1)** |
| **SERIESB.1** | 0.005424 (3) | 0.0321 (6) | 0.005424 (3) | 0.006143 (5) | **0.000104 (1)** | **0.000104 (1)** |
| **LACSTJRA.1** | 0.022971 (5) | 0.012493 (3) | 0.016164 (4) | 0.261073 (6) | **0.010394 (1)** | **0.010394 (1)** |
| **Ozone** | 0.004042 (4) | 0.03911 (5) | 0.00079 (2) | 0.131998 (6) | 0.005935 (3) | **0.000706 (1)** |
| **RAPPAHAN.1** | 0.006714 (3) | 0.014144 (5) | 0.006714 (3) | 0.048011 (6) | **0.001641 (1)** | **0.001641 (1)** |
| **HBCO.1** | 0.077075 (5) | 0.014459 (3) | 0.077075 (5) | 0.021279 (4) | **0.000486 (1)** | **0.000486 (1)** |
| **CURRENT.1** | **0.005603 (1)** | 0.007585 (2) | 0.031418 (5) | 0.050923 (6) | 0.010159 (4) | 0.00879 (3) |
| **ASKEW7.1** | 0.118959 (5) | **0.003655 (1)** | 0.118959 (5) | 0.080627 (4) | 0.03322 (3) | 0.012513 (2) |
| **SIMAR4.1** | 0.05864 (4) | 0.010436 (3) | 0.05864 (4) | 0.097847 (6) | **0.000093 (1)** | **0.000093 (1)** |
| **NYWATER.1** | **0.000024 (1)** | 0.008165 (3) | **0.000024 (1)** | 0.108533 (6) | 0.021238 (4) | 0.021238 (4) |
| **CONSUM.1** | 0.006701 (3) | 0.192148 (5) | 0.006701 (3) | 0.410414 (6) | 0.003062 (2) | **0.001941 (1)** |
| **children_per_woman** | **0.000027 (1)** | 0.408328 (5) | **0.000027 (1)** | 1.65767 (6) | 0.013115 (3) | 0.013115 (3) |
| **AMERICAN.1** | 0.099565 (4) | 0.006172 (3) | 0.099565 (4) | 0.27694 (6) | **0.000867 (1)** | **0.000867 (1)** |
| **SERIESJ.2** | **0.000311 (1)** | 0.021111 (3) | **0.000311 (1)** | 0.867712 (6) | 0.0531 (4) | 0.0531 (4) |
| **FRNCHB.1** | 0.031783 (4) | 0.103844 (2) | 0.032671 (4) | 0.235725 (6) | 0.013224 (3) | **0.008845 (1)** |
| **OTTER_L.1** | **0.000697 (1)** | 0.154446 (5) | 0.003607 (4) | 0.228931 (6) | 0.001445 (3) | 0.001383 (2) |
| **IBM.1** | 0.099506 (3) | 0.007062 (1) | 0.029243 (2) | 1.192122 (6) | 0.181934 (5) | 0.150301 (4) |
| **IV.1** | 0.053671 (4) | 0.01878 (3) | 0.053671 (4) | 0.479534 (6) | **0.000081 (1)** | **0.000081 (1)** |
| **HANKOU.1** | 0.005295 (4) | **0.000199 (1)** | 0.085691 (5) | 0.427704 (6) | 0.001 (2) | 0.001 (2) |
| **ESPANOLA.1** | 0.107704 (5) | 0.063331 (3) | 0.107704 (5) | 0.069497 (4) | 0.038676 (2) | **0.03053 (1)** |
| **NEUMUNAS.1** | 0.019752 (2) | 0.05087 (5) | 0.019752 (2) | 0.039879 (4) | 0.074946 (6) | **0.018099 (1)** |
| **NINEMILE.1** | 0.034372 (2) | 0.150897 (6) | 0.034372 (2) | 0.038648 (4) | 0.084953 (5) | **0.021218 (1)** |
| **NAVAJO.1** | 0.025011 (3) | 0.045677 (4) | 0.053738 (5) | 0.124633 (6) | **0.005094 (1)** | **0.005094 (1)** |
| **BLUME.1** | 0.002218 (2) | 0.107086 (6) | **0.001929 (1)** | 0.049793 (5) | 0.003823 (3) | 0.003519 (3) |
| **NILE2.1** | **0.00137 (1)** | 0.001152 (3) | **0.00137 (1)** | 0.193211 (6) | 0.054431 (5) | 0.014202 (4) |
| **LOGISTIC.1** | 0.080491 (3) | 0.200174 (6) | 0.080491 (3) | 0.16654 (5) | **0.000003 (1)** | **0.000003 (1)** |
| **Y.1** | 0.010648 (3) | 0.00421 (2) | 0.002022 (1) | 0.722437 (6) | 0.065461 (5) | 0.061986 (4) |
| **WBDELAWA.1** | 0.050875 (4) | 0.017499 (3) | 0.094832 (6) | 0.094832 (5) | **0.003033 (1)** | **0.003033 (1)** |
| **AMAZON.2** | 0.000339 (2) | 0.001309 (2) | **0.000156 (1)** | 0.219018 (6) | 0.051773 (4) | 0.051773 (4) |
| **CN.1** | 0.015912 | 0.079448 | 0.012808 | 0.007689 | 0.00342 | **0.00067 (1)** |
| **ASKEW14.1** | 0.021909 (4) | **0.000364 (1)** | 0.021909 (4) | 0.046782 (6) | 0.004908 (2) | 0.004908 (2) |
| **LACSTJIN.1** | **0.000059 (1)** | 0.043367 (4) | 0.029783 (5) | 0.094873 (6) | 0.001409 (2) | 0.001409 (2) |
| **CD.1** | **0.004928 (1)** | 0.139645 (5) | 0.041752 (4) | 0.476946 (6) | 0.010701 (2) | 0.010701 (2) |
| **FISHERT.1** | 0.023112 (3) | 0.117186 (6) | 0.033638 (4) | 0.08318 (5) | **0.008069 (1)** | **0.008069 (1)** |
| **RGNP.1** | 0.000071 (2) | 0.034112 (5) | **0.000009 (1)** | 0.079975 (6) | 0.01095 (4) | 0.00114 (3) |
| **AROSA.1** | **0.007575042 (1)** | 0.010614463 (2) | 0.242362866 (5) | 0.285212298 (6) | 0.01804633 (3) | 0.01804633 (3) |
| **U.1** | 0.005295366 (4) | **0.000199081 (1)** | 0.085691031 (5) | 0.427703736 (6) | 0.000999952 (2) | 0.000999952 (2) |
| **RACOON.1** | 0.001346976 (3) | 0.060859373 (5) | 0.001346976 (3) | 0.293946208 (6) | **0.000630665 (1)** | **0.000630665 (1)** |
| **BND.1** | 0.00047181 | 0.111166889 | **0.0000889 (1)** | 0.043800188 | 0.002446093 | 0.000256424 (2) |
| **Average** | 0.0065 (2.4693) | 0.0158 (3.3663) | 0.0071 (2.5742) | 0.0351 (4.5742) | 0.00463 (2.8019) | **0.00256 (2.0571)** |
| **STD** | 0.028074 | 0.063828 | 0.043792 | 0.274294 | 0.028135 | 0.020976 |

## J.2 STATISTICAL SIGNIFICANCE OF AUTOFORECAST

We now show the pairwise statistical test results between every pair of methods by Wilcoxon signed rank test in Table 15. Statistically better method shown in **bold** (both marked bold if no significance). We re-emphasize that we use the pairwise Wilcoxon signed rank test on performances (MSE) across datasets (significance level $p < 0.05$). In the left, univariate testbed is shown where AUTOFORECAST is statistically significantly better than most baselines including GB, AS, and AUTOFORECAST-TSL. In the right, multivariate testbed is shown where AUTOFORECAST is statistically significantly better than AS, MLP, and AUTOFORECAST-TSL.

We next turn our attention to the tuning of the time-series meta-learner and the timing of AUTOFORECAST, respectively. In particular, we show both the computational cost and the inference time for AUTOFORECAST and compare it with our different baselines.

Table 13: Method evaluation in multivariate testbed (average MSE). The most performing (lowest MSE) method is highlighted in **bold**. The rank is provided in parenthesis (lower ranks denote better performance). AUTOFORECAST achieves the best average MSE and average rank among all meta-learners. In particular, it has the best performance (lowest average MSE) on 28 datasets out of the 40 multivariate datasets and has comparable performance for remaining datasets.

| Dataset | Global Best | AS | ISAC | MLP | AUTOFORECAST-TSL | AUTOFORECAST |
|---|---|---|---|---|---|---|
| **Processed_S&P** | 0.137347 (2) | 0.455675 (5) | 0.137347 (2) | 4.936163 (6) | 0.158537 (4) | **0.071178 (1)** |
| **ozone_onehr** | 0.780519 (5) | 0.364615 (3) | 0.780519 (5) | 0.765257 (4) | 0.189151 (2) | **0.042093 (1)** |
| **Occ_Train** | **0.000106 (1)** | 0.66462 (5) | **0.000106 (1)** | 0.717224 6) | 0.303732 (4) | 0.072843 (3) |
| **Scanline** | 0.00811 (2) | 0.627556 (5) | 0.00811 (2) | 1.292029 (6) | 0.064343 (4) | **0.005702 (1)** |
| **knoy_mpu_3_300** | **0.009803 (1)** | 0.176418 (6) | **0.009803 (1)** | 0.04145 (3) | 0.10648 (5) | 0.043316 (4) |
| **energydata_complete** | 0.273006 (3) | 0.298254 (5) | 0.273006 (3) | 3.02911 (6) | 0.145331 (2) | **0.065071 (1)** |
| **Adobe_Service_CPU_Mem_15d** | 0.264722 (2) | 0.813373 (5) | 0.264722 (2) | 10.782728 (6) | 0.581017 (4) | **0.120063 (1)** |
| **knoy_mpu_1_340** | 0.009342 (4) | 0.005052 (3) | 0.009342 (4) | 0.239649 (6) | 0.001921 (2) | **0.000374 (1)** |
| **iowa-electricity** | 0.003171 (3) | 0.025284 (5) | 0.003171 (3) | 0.966798 (6) | **0.000001 (1)** | **0.000001 (1)** |
| **knoy_mpu_1_400** | 0.077215 (4) | 0.020974 (3) | 0.077215 (4) | 0.101848 (6) | **0.000009 (1)** | **0.000009 (1)** |
| **knoy_mpu_2_400** | 0.132552 (4) | **0.005615 (1)** | 0.132552 (4) | 2.452516 (6) | 0.011195 (3) | 0.00656 (2) |
| **Occupancy** | **0.000097 (1)** | 0.056036 (4) | **0.000097 (1)** | 2.821039 (6) | 0.253453 (5) | 0.022591 (3) |
| **ozone_eighthr** | 0.131161 (2) | 0.286562 (5) | 0.131161 (2) | 4.974266 (6) | 0.151916 (4) | **0.090387 (1)** |
| **knoy_mpu_3_400** | 0.024336 (4) | 0.005892 (3) | 0.024336 (4) | 0.083823 (6) | 0.001408 (2) | **0.000618 (1)** |
| **quality_control** | 0.028198 (4) | 0.027566 (3) | 0.028198 (4) | 4.303693 (6) | 0.025724 (2) | **0.006722 (1)** |
| **knoy_mpu_1_500** | 0.022832 (4) | 0.015874 (3) | 0.022832 (4) | 0.063941 (6) | 0.000024 (2) | **0.000005 (1)** |
| **knoy_mpu_3_100** | **0.008206 (1)** | 0.014191 (3) | **0.008206 (1)** | 0.021298 (4) | 0.025675 (5) | 0.025675 (5) |
| **knoy_mpu_1_360** | 0.023268 (5) | 0.018886 (4) | 0.023268 (5) | 0.001753 (3) | 0.001025 (2) | **0.000034 (1)** |
| **knoy_mpu_2_100** | 0.005647 (3) | 0.009183 (5) | 0.005647 (3) | 1.82924 (6) | **0.000588 (1)** | **0.000588 (1)** |
| **knoy_mpu_2_500** | 0.042863 (3) | 0.450295 (5) | 0.042863 (3) | 0.703812 (6) | 0.042598 (2) | **0.001608 (1)** |
| **Sales_Transactions** | 0.068063 (2) | 0.122557 (5) | 0.068063 (2) | 0.57930 (6) | 0.074798 (4) | **0.005779 (1)** |
| **co2-concentration** | **0.001174 (1)** | 0.044724 (5) | **0.001174 (1)** | 0.613426 (6) | 0.006403 (4) | 0.001976 (3) |
| **knoy_mpu_1_100** | 0.00652 (2) | 0.123459 (6) | 0.00652 (2) | 0.064915 (5) | 0.006806 (4) | **0.002683 (1)** |
| **Processed_NASD** | 0.037896 (2) | 0.255123 (5) | 0.037896 (2) | 0.6712377 (6) | 0.117047 (4) | **0.020919 (1)** |
| **knoy_mpu_3_380** | 0.081679 (3) | 0.103497 (5) | 0.081679 (3) | 0.627697 (6) | 0.023772 (2) | **0.005902 (1)** |
| **knoy_mpu_2_320** | 0.036278 (3) | 0.014435 (2) | 0.036278 (3) | 0.193747 (6) | 0.069011 (5) | **0.01057 (1)** |
| **knoy_mpu_2_380** | **0.000959 (1)** | 0.009084 (4) | **0.000959 (1)** | 1.172323 (6) | 0.025441 (5) | 0.001284 (3) |
| **us-employment** | **0.005489 (1)** | 0.578479 (5) | **0.005489 (1)** | 4.724258 (6) | 0.046471 (4) | 0.009717 (3) |
| **knoy_mpu_1_380** | 0.052463 (4) | 0.131345 (6) | 0.052463 (4) | 0.000476 (2) | 0.008725 (3) | **0.000013 (1)** |
| **knoy_mpu_2_200** | 0.021158 (3) | 0.021852 (5) | 0.021158 (3) | 0.692687 (6) | 0.002585 (2) | **0.000488 (1)** |
| **fast-storage-1** | 0.113182 (2) | 0.69159 (5) | 0.113182 (2) | 2.765069 (6) | 0.130727 (4) | **0.032013 (1)** |
| **knoy_mpu_1_600** | 0.103141 (5) | 0.014413 (3) | 0.103141 (5) | 0.034072 (4) | 0.002585 (2) | **0.000359 (1)** |
| **knoy_mpu_3_200** | 0.011019 (4) | 0.006598 (3) | 0.011019 (4) | 0.463735 (6) | 0.002928 (2) | **0.002038 (1)** |
| **Processed_DJI** | **0.002731 (1)** | 0.407639 (5) | **0.002731 (1)** | 6.420626 (6) | 0.110294 (4) | 0.027849 (3) |
| **Processed_RUSS** | 0.109793 (3) | 0.159702 (5) | 0.109793 (3) | 2.493557 (6) | 0.072741 (2) | **0.026686 (1)** |
| **knoy_mpu_3_600** | 0.006719 (3) | 0.024021 (6) | 0.006719 (3) | **0.000626 (1)** | 0.011989 (5) | 0.003086 (2) |
| **knoy_mpu_2_600** | 0.072972 (4) | 0.031156 (3) | 0.072972 (4) | 1.704111 (6) | **0.005216 (1)** | **0.005216 (1)** |
| **Processed_NYSE** | 0.099705 (3) | 0.50898 (5) | 0.099705 (3) | 2.794788 (6) | 0.073811 (2) | **0.029246 (1)** |
| **Average** | 0.0584 (2.3191) | 0.1683 (3.0851) | 0.0584 (2.3191) | 1.6938 (3.8723) | 0.065 (2.3404) | **0.0163 (1.3191)** |
| **STD** | 0.1252 | 0.2295 | 0.1252 | 2.3421 | 0.1068 | 0.0271 |

Table 14: Results for one-step ahead forecasting (Average MSE across all datasets) for both testbeds. The seven SOTA forecasting models have worse performance (higher MSE) compared to most of the meta-learners. For each SOTA, we chose the model with the best average performance from all model variants. Moreover, AUTOFORECAST has the best performance for both testbeds.

| Testbed | Seasonal Naive | DeepAR | Deep Factors | Random Forest | Prophet | Gaussian Process | VAR |
|---|---|---|---|---|---|---|---|
| **Univariate** | 0.0345 | 0.0164 | 0.0217 | 0.0199 | 0.0155 | 0.1661 | 0.0602 |
| **Multivariate** | 0.0149 | 0.0085 | 0.0135 | 0.0071 | 0.0065 | 0.2576 | 0.9865 |
| | Global Best | AS | ISAC | MLP | AUTOFORECAST-TSL | AUTOFORECAST | |
| **Univariate** | 0.0065 | 0.0158 | 0.0071 | 0.0351 | 0.00463 | **0.00256** | |
| **Multivariate** | 0.0050 | 0.0139 | 0.0046 | 0.0121 | 0.00541 | **0.00124** | |

Table 15: Pairwise statistical test results between every pair of methods by Wilcoxon signed rank test. Statistically better method shown in **bold** (both marked bold if no significance). In the left, univariate testbed is shown where AUTOFORECAST is statistically significantly better than GB, AS, and AUTOFORECAST-TSL. In the right, multivariate testbed is shown where AUTOFORECAST is statistically significantly better than AS, MLP, and AUTOFORECAST-TSL.

| Ours | Baseline | p-value | | Ours | Baseline | p-value |
|---|---|---|---|---|---|---|
| **AUTOFORECAST** | GB | $9.0712 \times 10^{-5}$ | | **AUTOFORECAST** | **GB** | 1.0 |
| **AUTOFORECAST** | AS | $1.0726 \times 10^{-37}$ | | **AUTOFORECAST** | AS | $3.9399 \times 10^{-7}$ |
| **AUTOFORECAST** | **ISAC** | 0.1349 | | **AUTOFORECAST** | **ISAC** | 0.8240 |
| **AUTOFORECAST** | **MLP** | 0.0657 | | **AUTOFORECAST** | MLP | 0.0004 |
| **AUTOFORECAST** | AUTOFORECAST-TSL | $8.1683 \times 10^{-15}$ | | **AUTOFORECAST** | AUTOFORECAST-TSL | 0.0025 |
| **AUTOFORECAST-TSL** | GB | $2.2611 \times 10^{-8}$ | | **AUTOFORECAST-TSL** | GB | 0.00254 |
| **AUTOFORECAST-TSL** | AS | $1.5760 \times 10^{-14}$ | | **AUTOFORECAST-TSL** | AS | $5.8013 \times 10^{-7}$ |
| **AUTOFORECAST-TSL** | ISAC | $2.3843 \times 10^{-16}$ | | **AUTOFORECAST-TSL** | ISAC | $1.5598 \times 10^{-5}$ |
| **AUTOFORECAST-TSL** | MLP | $1.1658 \times 10^{-26}$ | | **AUTOFORECAST-TSL** | MLP | $3.4572 \times 10^{-8}$ |
| **GB** | AS | $9.4952 \times 10^{-33}$ | | **GB** | AS | $3.9399 \times 10^{-7}$ |
| **GB** | ISAC | 0.0322 | | **GB** | **ISAC** | 0.8240 |
| **GB** | MLP | $4.5489 \times 10^{-9}$ | | **GB** | MLP | 0.0004 |
| **AS** | ISAC | $1.7842 \times 10^{-37}$ | | **AS** | ISAC | $1.4217 \times 10^{-7}$ |
| **AS** | MLP | $4.4658 \times 10^{-54}$ | | **AS** | MLP | $6.6612 \times 10^{-8}$ |
| **ISAC** | MLP | $2.2062 \times 10^{-31}$ | | **ISAC** | MLP | $3.7789 \times 10^{-8}$ |

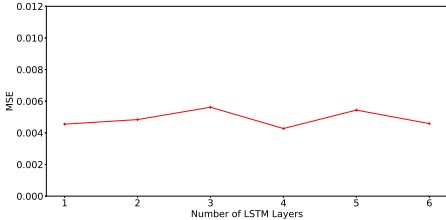

Figure 9: The performance of the time-series meta-learner $\Theta$ vs. number of the LSTM layers of $\Theta$ for the univariate testbed. LSTM with 4 layers gives the best performance (lowest MSE).

Figure 10: The performance of the time-series meta-learner $\Theta$ vs. number of units per layer of $\Theta$ for the univariate testbed. LSTM with 50 units per layer gives the best performance.

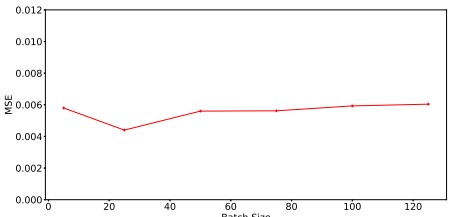
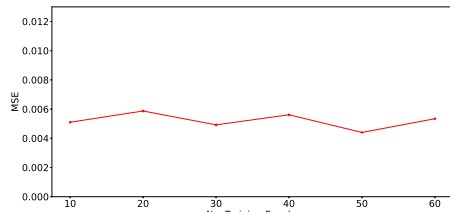

Figure 11: The performance of the time-series meta-learner $\Theta$ vs. the training batch size of $\Theta$ for the univariate testbed. A batch with size = 25 instances gives the best performance.

Figure 12: The performance of the time-series meta-learner $\Theta$ vs. number of training epochs of $\Theta$ for the univariate testbed. Training with 50 epochs gives the best performance.

## J.3 TUNING OF TIME-SERIES META-LEARNER $\Theta$

We show the effect of different hyper-parameters used in the training of the time-series meta-learner $\Theta$ on the performance (in terms of MSE under the selected model by $\Theta$). For searching on each parameter, we fix the other parameters on their best values. In the interest of space, we show full details for the univariate testbed. Figure 9 shows the effect of the number of the LSTM layers of the time-series meta-learner. We observe that LSTM with 4 layers gives the best performance (lowest MSE). Second, Figure 10 shows the effect of the number of the units in the LSTM layer of the time-series meta-learner. We observe that LSTM with 50 units per layer gives the best performance (lowest MSE). Third, Figure 11 shows the effect of the training batch size of the time-series meta-learner. A batch with size = 25 instances gives the best performance (lowest MSE). Finally, Figure 12 illustrates the effect of the number of training epochs of the time-series meta-learner. We note that training with 50 epochs gives the best performance (lowest MSE). We have also used dropout rate of 0.2 to prevent over-fitting.

## J.4 COMPARING COMPUTATIONAL COST OF AUTOFORECAST

Table 16 shows that AUTOFORECAST has comparable computational training cost compared to the other meta-learners baselines. We re-emphasize that the running time of such offline training phase is less critical since it is done only once. However, this experiment shows that our better time series model selection performance does not entail a prohibitive training cost.

Table 16: Computational cost for training (in seconds) for both univariate and multivariate testbeds. AUTOFORECAST has comparable computational cost compared to other meta-learners baselines.

| Dataset Testbed | Global Best | AS | ISAC | MLP | AUTOFORECAST-TSL | AUTOFORECAST |
|---|---|---|---|---|---|---|
| **Univariate** | N/A | $308.9301 \pm 46.1968$ | $278.8083 \pm 57.9900$ | $705.2908 \pm 123.3715$ | $334.8091 \pm 31.6808$ | $670.5855 \pm 31.5465$ |
| **Multivariate** | N/A | $194.3877 \pm 39.7441$ | $182.4753 \pm 34.3238$ | $411.9337 \pm 41.7406$ | $178.1978 \pm 18.0098$ | $376.3956 \pm 40.0195$ |

## J.5 INFERENCE RUN TIME STATISTICS

**Comparison between** AUTOFORECAST **and naïve approach:** We compare the inference time that AUTOFORECAST versus the inference time of the naïve approach, i.e., doing inference over all possible models. We first show the aggregate statistics for the gain across all datasets for each testbed. Figure 3 illustrates the reduction in inference time for using AUTOFORECAST over the naïve approach for both univariate and multivariate testbeds. In particular, AUTOFORECAST gives a median gain of at least 42X over the naïve approach for both testbeds.

**Dataset-wise inference time comparison:** Next, we pick several random groups of 10 datasets each and show the time (in seconds) that AUTOFORECAST takes versus the time the naïve approach takes for forecasting model selection. Figures 13-14 shows such comparison for the univariate testbed. It is noted the higher reduction of inference time for larger datasets (i.e., with more data points).

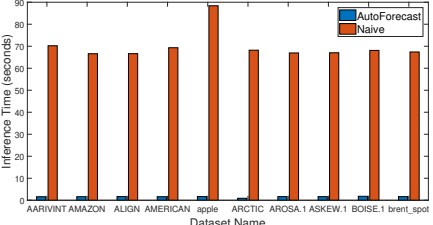
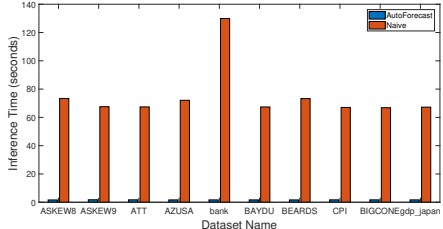

Figure 13: Inference time comparison between AUTOFORECAST and naïve approach.

Figure 14: Inference time comparison between AUTOFORECAST and naïve approach.

**Multivariate testbed:** We also observe that AUTOFORECAST (meta-feature generation and model selection) takes less than 1.2 second on most time series datasets for multivariate testbed.

**Time overhead of** AUTOFORECAST **relative to training of selected model:** Figure 15 shows that it incurs only negligible overhead relative to actual training of the selected model (with median = 0.1%). Similarly, AUTOFORECAST incurs only negligible overhead relative to actual training of the selected model for the multivariate testbed (with median = 0.3%) (Figure is omitted for multivariate testbed). This shows that AUTOFORECAST is lightweight, incurring small selection time overhead.

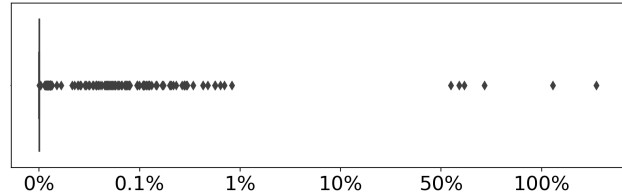

Figure 15: Boxplot of time AUTOFORECAST takes relative to training of selected model in univariate testbed. AUTOFORECAST incurs negligible overhead, (median = 0.1%).

## K INTERPRETABLE FEATURES OF DATASETS

Now, we show some of the interpretable input feature values for all datasets in the univariate testbed. In particular, we show mean, median, variance, skeweness, kutosis, absolute energy, and benford correlation for each of the individual time series in that testbed. Such features are shown in Figures 16-23. Note that time-series datasets have standard normalization before feature extraction. Such features values show the diversity in the datasets, where each dataset has different characteristics. We emphasize that in the interest of space, we show sample of these interpretable features.

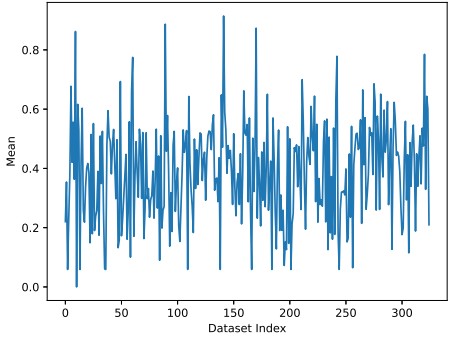

Figure 16: Mean

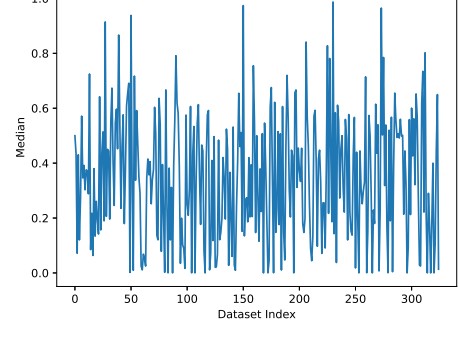

Figure 17: Median

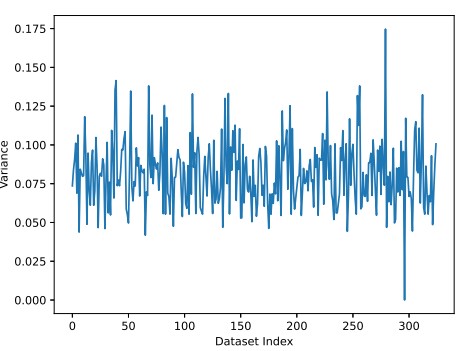

Figure 18: Variance

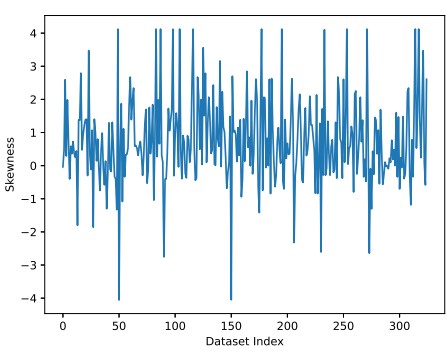

Figure 19: Skeweness

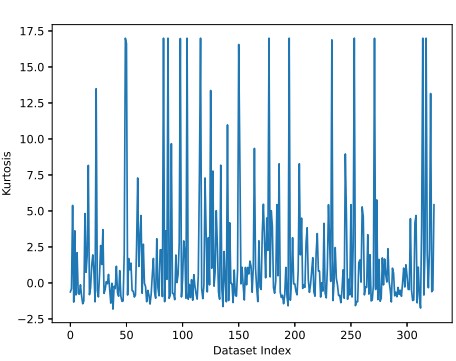

Figure 20: Kurtosis

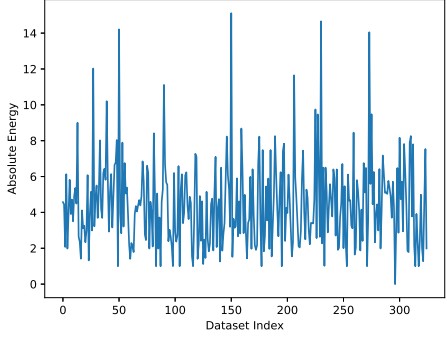

Figure 21: Absolute Energy

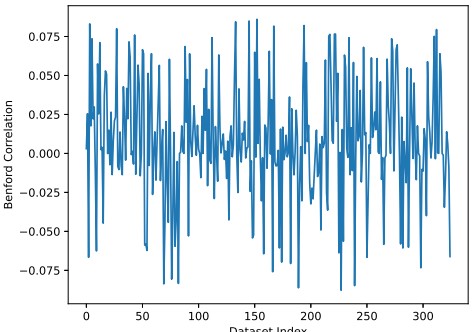

Figure 22: Benford Correlation



Figure 23: Existence of duplicates (binary)

