# OpenReview forum: "Automatic Forecasting via Meta-Learning"
_ICLR.cc/2022/Conference — ICLR 2022 Submitted_

### Official Review · Reviewer_sRFp · 2021-11-01

**Correctness:** 3
**Technical Novelty And Significance:** 2
**Empirical Novelty And Significance:** 2
**Recommendation:** 3
**Confidence:** 3

**Main Review:**

As a caveat, I'm not an expert in meta-learning, so I cannot comment expertly on the contributions of the paper in the meta-learning space.

Overall, the paper proposes an interesting and practically relevant problem. Unfortunately, I found it hard to read from a language point of view and I'm missing important details without which I have a hard time to follow the paper. For example: what actually is a model in the way that the authors use it? I understand that the hyperparameters are fixed (so each M_i in the model "space" has different hyper parameters but may be from the same model family) but are the weights/parameters as well? If the hyperparameters are fixed, then the meta-learner basically is a combination of model-selector with HPO -- in this case, we should probably compare to an HPO method in the experiments (I understand that typically HPO methods wouldn't be able to produce results "quick" enough, but they would provide a natural upper bound on performance).

If weights/parameters are not fixed in each M_i not, where are they re-trained? Or are they fine-tuned only? How do we transfer a trained model from one forecasting task to another when important details differ (e.g., length of the forecast horizon, length of the history)? What actually is a forecasting task in the authors' understanding?

I understand that space is limited and not all of this can be mentioned, but then, there are details offered about LSTMs which can be considered standard and shorted (on page 5).

detailed review/questions:
- page 1: in what sense is the burden "infeasible"? Is it computationally too expensive? Further onwards, one desideratum is that the selection should be "quick" -- isn't a even more important feature that the selected method is accurate?

- it's unclear to me how you run forecasting models on different length forecast horizons w_t in the meta-learning training data set. How are these models trained? Some of those forecasting models that you employ required a fixed-length horizon.

- page 3: v_i is the number of *variables* in D_i. Variables is an odd name here. Probably "number of time series" or "items" is more appropriate. Or am I missing something?

- why is the model space called a "space"? Isn't this just a "set"?

- Also, in terms of exposition, it's unclear to me what exactly a model is until Definition 3 -- but by then this has already been extensively used. So, I would ask to move Definition 3 ahead of the other definition. As it, I'm left wondering in Definition 2 whether a "model" is a fully specified model (e.g., the weights in the neural network are already fixed? are hyper-parameters fixed)? In Definition 2, it's implicit that it's a fully trained model, but that's not what seems to be written in Definition 3. It's also unclear to me what a forecasting "algorithm" is. A "method" would make some sense to me, but "algorithm" evokes associations around which algorithm to use when estimating the free parameters of the model -- surely that isn't meant here.

- why are the covariates of a data set called meta-features? That may be standard meta-learning terminology, but to me it reads odd. You're simply extracting features from the time series, I can't seem to find the "meta" nature here (apart for that the meta-learner uses them?) Also, there are standard time series features like tsfresh, catch22 around or funkier things by Francheschi et all (NeurIPS 2019) which should at least be mentioned.

- page 5: paragraph starting "General Meta-learner..." what does from *running* all models actually mean? Are you re-training then doing inference? This is a bit too condensed/sloppily written for me to actually understand what's going on.

- paragraph starting "LSTM-based Time-series..." I don't understand how we construct F^i_j and p^i_j when on page 3 it sounded as if the forecasting windows w_t are sampled randomly (random time points and random lengths -- I also don't understand how we train the models to arrive at the performance tensors by the way -- do we re-train for every different w_t?). I'm missing the connect between this random-looking construct of w_t and the systematically constructed F^i_j and p^i_j . Also, why is this called a "multi-regression" model? Isn't this just a forecasting model again where we have co-variates (F) available?

- page 6: seasonal naive is surely not a SOTA method, but rather a baseline. Overall, the selection of methods makes sense to me, but not for the reason (being SOTA) listed there. Instead, they represent a large selection of methods available for which I would laude the authors. If state of the art was the primary reason, a transformer-based method (TFT for example) and NBEATS should also complement DeepAR. As far as I know GluonTS includes some of these models in reasonable implementations.

- it would be helpful to define clearly what the characteristics of a "forecasting task" exactly are. I assume: length of the forecasting horizon/window, maybe also length of history.

- the data set selection for training the meta-learners doesn't contain some standard forecasting benchmark data sets (e.g., M4, M5) which are collected in the Monash repository (which appeared concurrently with the present paper). This selection by the authors should be justified further. As is, it seems low in the number of time series overall, in particular when deep learning based methods are used for which one would expect a larger data hunger. Adobe real traces (e.g., Appendix G2) are mentioned, but there isn't a pointer to literature for them. If the authors are publishing this data set for the first time, this should be mentioned more clearly as a contribution.

- Appendix A, Figure 4. I understand what the authors are trying to achieve here, but as is, the figure doesn't help my understanding. For example, some further grouping would make it easier. Where on the x-axis are the random forest models? Maybe there's not a single dominant model (again, the authors need to define more clearly what a model is), but maybe a model family?

**Summary Of The Paper:**

This paper present a novel approach to addressing a fundamental problem in forecasting research: given the plethora of methods available and no clearly superior approach overall, which forecasting method should be chosen for a new dataset/forecasting task combination. The authors propose a combination of a static (not taking the time evolution of accuracy into account) meta-learning model with one that takes the evolution of accuracy over time into account. They apply this methodology using a number of well-chosen forecasting methods on public data sets. Some of these (e.g., Adobe traces), I hadn't see before, and their discovery may be an important contribution in itself.

**Summary Of The Review:**

While I think that the problem that the authors address in this paper is of great practical importance and has plenty of scientific challenges which attention, I cannot recommend the present paper for acceptance. I base this primarily on the exposition of the work which leaves too many too important details open. This inhibits my understand of the paper (and the appendices do not shed light on those details when I checked for it).

---

### Official Review · Reviewer_cQ6U · 2021-11-02

**Correctness:** 2
**Technical Novelty And Significance:** 2
**Empirical Novelty And Significance:** 2
**Recommendation:** 3
**Confidence:** 4

**Main Review:**

Once the meta dataset has been computed, the problem
basically becomes a cost-sensitive classification problem,
where the goal is to predict the right forecasting model (class)
where wrong ones are penalized according to how much
excess loss they will cause. But none of the baselines is
following this simple setup.

The baselines seem to be very weak: they are consistently
worse than the constant model that always predicts the same
forecasting model. More challing baselines would make
the experiments more interesting.

The notation of the paper is not easy to follow.
- It is not clearly defined what the windows are (beginning of
  sect. 3).
- Why are the metafeatures F_t^i in \R^{d\times v_i} (before
  eq. 3), i.e., we have d metafeatures per channel (v_i)?
  Some of the features in table 8 seem not to depend on
  the channel, e.g., window length and landmark features.
  - If they depend on the number of channels v_i, how can
    the regression model then generalize over different datasets
    with different numbers of channels?
- eq.2: what is "w_i" ?


**Summary Of The Paper:**

The authors address the problem of selecting the best
forecasting model for a time series dataset. Datasets
are described by a large vector of meta features, then
two regression models are learnt to predict the test loss
of a forecasting model for such a dataset at a given
cutoff time point (called window): one predicts directly
on the meta features computed until the cutoff time point,
the other predicts autoregressively starting from the
first cutoff timepoint until the target cutoff time
point is reached. In experiments on two meta datasets
(called performance tensors), one univariate, one multivariate,
it is shown that esp. the autoregressive model outperforms
the best constant model (called global best) as well as
some baselines.


**Summary Of The Review:**

strengths:
s1. interesting set of t.s. metafeatures.
s2. interesting meta dataset for t.s. forecasting.

weaknesses:
w1. problem is basically a cost-sensitive classification problem,
    but the loss does not reflect this.
w2. baselines seem to be very weak.
w3. complicated notation.

---

### Official Review · Reviewer_M5ZB · 2021-11-02

**Correctness:** 2
**Technical Novelty And Significance:** 3
**Empirical Novelty And Significance:** 4
**Recommendation:** 5
**Confidence:** 3

**Main Review:**

**Strengths**
- This paper introduces a novel approach for meta learning for time series datasets. While meta learning has been widely studied for various types of datasets, it has not been properly applied to time series. This paper takes one of the first steps in applying meta learning to time series.
- While there exist a multitude of time series datasets, it is an immense effort to collect all of them together under a single corpus. This paper introduces a corpus constructed from Kaggle and other open source repositories and provide it as a part of the supplementary material.
- Extensive experimental evaluation shows improved performance of the proposed algorithm over other meta-learning baselines.

**Weaknesses**
- Meta-learning performance can often be affected by the diversity of the training datasets. Yet, there is no discussion about the diversity of the time series in the corpus. It will be interesting to see some of the interpretable input feature values (for example, window length, mean, variance, etc) for each of the individual time series (or aggregated statistics).
- The sources of the collected datasets are also not mentioned. It is important that meta learning datasets be collected from a diverse set of sources in order to be useful. While there is a reasonable amount of diversity within Kaggle datasets, Adobe datasets maybe quite similar to each other.
- Some of the experimental details and results are unclear or inconsistent.
  - Page 7: Hit-at-k is not clearly defined.
  - Page 8: The p-values right below Table 4 do not match the numbers in Table 5. Similarly for the numbers in Page 9 right below Table 6. The meaning of these numbers is not clear.
  - Table 4 presents the mean MSE over all datasets. However, the datasets can be quite diverse in scales as is also reflected in the variances (which are larger than the mean values itself). A normalized MSE metric seems more apt as a metric in this case.
  - Table 5 results are not strong enough. However, I do not see this as a major drawback of the paper.

## Update after rebuttal:
Thanks for the response. The diversity of the datasets is unfortunately not clear from the paper. The presentation of the paper needs to be improved greatly. In particular Tables 10 and 11 are unreadable. Moreover, I count only 23 multivariate datasets from Table 11. It can be much more readable if instead of the raw time series IDs, they are actually organized into categories (like finance, energy, etc). While I don't mind having the variable names in the table as well, a shorter table summarizing the diversity in the data will be much more useful. While this paper addresses an interesting problem, it needs significant improvements. After reading the other reviews and responses, I will be keeping the same score.

**Summary Of The Paper:**

This paper proposes a meta learning approach for time series. Specifically, the approach learns to select the best model and hyper-parameters for unseen test datasets, when trained on performance data of various models applied to training datasets. The approach utilizes various dataset specific features in order to make the predictions. Experimental results show an improved performance over other meta learning baselines. The corpus of time series datasets that was used for the experiments, is also provided as supplementary material.

**Summary Of The Review:**

This paper makes some significant contribution towards a meta learning approach for time series forecasting, including releasing a dataset. However, due the issues in the experimental evaluation/results, and the dataset collection process, I am leaning towards a reject. I am willing to increase my score if my concerns are satisfactorily addressed.

---

> ### Author Response · Authors · 2021-11-13
> **Response to Reviewer 2 (M5ZB)**
>
> We thank the reviewer for the positive feedback and insightful comments. Below we reply to the questions the reviewer raised.
>
> Comment: Meta-learning performance can often be affected by the diversity of the training datasets. Yet, there is no discussion about the diversity of the time series in the corpus. It will be interesting to see some of the interpretable input feature values (for example, window length, mean, variance, etc.) for each of the individual time series (or aggregated statistics).
>
> Response: Thanks for the comment. The diversity aspect of our datasets is addressed in the next comment's response. We emphasize that we have released the meta-features matrices for each dataset in our supplementary material along with the datasets. We have also added interpretable features (mean, variance, median, absolute energy, correlation, kurtosis, and skewness)  for each individual time series of the datasets in Appendix K, thanks for the suggestion.
>
> Comment: The sources of the collected datasets are also not mentioned. It is important that meta-learning datasets be collected from a diverse set of sources in order to be useful. While there is a reasonable amount of diversity within Kaggle datasets, Adobe datasets may be quite similar to each other.
>
> Response: Thanks for the comment. We acknowledge that diversity of sources makes the meta-learning model learn from such diversity since model selection on test data can thus benefit from the prior experience on that diverse train set. For that purpose, we use benchmark datasets from Kaggle, Adobe real traces, and other open-source repositories, where most of the datasets are from different application domains (e.g., finance, IoT, energy, storage, etc.). We have released these benchmark datasets to help the community build on our work. In particular, the Adobe trace datasets record CPU and Memory usage for 50 different services running in Adobe production clusters collected for 15 days from May 1 to May 15, 2021. Further, notice that Adobe datasets are only two datasets of our 40 multivariate datasets in the multivariate testbed (i.e., the remaining 38 multivariate datasets are from diverse sources and the 308 univariate datasets are from different sources too). We also emphasize that in our multivariate testbed, each dataset can have a different type. For instance, the Adobe service CPU and memory 15 days utilization dataset is a "Homogeneous" multivariate dataset (i.e., has 50 time series for CPU and 50 time series for Memory from different ML traces measuring the same variable), however, the rest of multivariate datasets are heterogeneous (i.e., measuring different variables). Adobe datasets are shared within our supplementary material folder. We expanded Appendix G.1 and G.2 for clarifying this issue.
>
> Comment: Some of the experimental details and results are unclear or inconsistent.
>
> (a) Page 7: Hit-at-k is not clearly defined.
>
> Response: Thanks for pointing this, we have modified the Hit-at-k definition on page 7: "Hit-at-k accuracy which indicates whether the selected model by our algorithm (AUTOFORECAST) is within the actual top-K models" (e.g., with 300 datasets, Hit-at-10 accuracy of 21% indicates that our selected model lies within the top-10 models in 63 datasets of the 300 datasets).
>
> (b) Page 8: The p-values right below Table 4 do not match the numbers in Table 5. Similarly for the numbers on Page 9 right below Table 6. The meaning of these numbers is not clear.
>
> Response: Sorry for the confusion, both of these values in parenthesis below Table 4 and Table 6 are MSE values, not p-values. We replaced them with the p-values in the revised paper on page 8.
>
> (c) Table 4 presents the mean MSE over all datasets. However, the datasets can be quite diverse in scales as is also reflected in the variances (which are larger than the mean values itself). A normalized MSE metric seems more apt as a metric in this case.
>
> Response: We agree that datasets can be diverse. Therefore, we had presented (in our original submission) the MSE-per-dataset (data-wise MSE) in Table 12 (for one fold of Uni-variate datasets) and Table 13 (for all multivariate datasets). We plan to add the normalized MSE metric as well in the final version.
>
> Kindly let us know if our response addresses your concerns. We will be happy to answer if there are additional issues/questions.
>
> Sincerely,
>
> Authors of Paper 3114

---

### Official Review · Reviewer_ZkNj · 2021-11-04

**Correctness:** 3
**Technical Novelty And Significance:** 3
**Empirical Novelty And Significance:** 3
**Recommendation:** 6
**Confidence:** 3

**Main Review:**

## Strengths
- The overall problem and method are novel and interesting. Model selection for time-series forecasting is a new setting and the proposed AUTOFORECAST is reasonable to solve the problem in an efficient and effective way.
- The constructed benchmark is large and comprehensive, which serves as a good testbed for the proposed problem and method.

## Weaknesses
- Several modules are named with "meta", but it is unclear to me where the meta procedure happens, e.g. inner-loop, outer-loop.
- The overall formulation and writing are somehow hard to follow. Since this is a relatively new problem, clear formulation and problem definition are important for the audience.
- It seems the baselines in Sec 5.2 are relatively old.

### Detailed Comments
As mentioned above, I am not aware why the modules should be named "meta-xxx". There is no meta-learning procedure applied to learn these modules. For meta-learning, during training, a meta-model and meta-parameters are learned in a bi-level optimization manner. During testing, the task-specific parameters can be adapted to perform the target tasks.
Figure 1 is hard to see clearly without extensive zoom-in.
There are several meta-learning works for time-series forecasting missing, e.g. [R1, R2].


[R1] Pan, Zheyi, et al. "Spatio-temporal meta learning for urban traffic prediction." IEEE Transactions on Knowledge and Data Engineering (2020).
[R2] Narwariya, Jyoti, et al. "Meta-learning for few-shot time series classification." Proceedings of the 7th ACM IKDD CoDS and 25th COMAD. 2020. 28-36.

**Summary Of The Paper:**

This paper formulates the problem of automatic and fast selection of the best time-series forecasting model as a meta-learning problem.
Specifically, an AUTOFORECAST model consisting of a temporal meta-learner and a general meta-learner are proposed.

A large benchmark and a collection of models are established to evaluate the problem.


**Summary Of The Review:**

Overall, I think the proposed problem is novel and new. The proposed method is reasonable to solve the problem.
However, there are issues that need to be addressed.

---

> ### Author Response · Authors · 2021-11-12
> **Response to Reviewer 1 (ZkNj)**
>
> We thank the reviewer for the positive feedback and insightful comments. Below we reply to the questions the reviewer raised.
>
> Question 1: Several modules are named with "meta", but it is unclear to me where the meta procedure happens, e.g. inner-loop, outer-loop. The overall formulation and writing are somehow hard to follow. Since this is a relatively new problem, clear formulation and problem definition are important for the audience.
>
> Response: Thanks for the comment, we emphasize that we use the term "meta-learning'' in the context of the traditional principle of meta-learning which is building upon prior experience on a set of historical tasks to "do better'' on a new task. We build the experience across different datasets using our “general meta-learner" and build experience on the sequential dependence among the same dataset using the "LSTM time-series meta-learner''. We also capture task similarity between a new input task (dataset) and historical datasets using the "meta-features''. We also emphasize that our proposed method is faster compared to gradient-descent-based bi-level meta-learners, e.g., on our univariate testbed N-Beats has significantly slower training (average = 3600 seconds) and inference time (average = 101 seconds) compared to \name (average = 670 seconds for training and 1.13 seconds for inference). We added such a discussion on Appendix I. We also tried to make the formulation more clear based on your and other reviewers' comments.
>
>
> Question 2: It seems the baselines in Sec 5.2 are relatively old.
>
> Response. We acknowledge that more baselines make the paper stronger. While we cannot really run more baselines and update all tables during the course of the discussion period, there are two points to clarify regarding our baselines: First, since our proposed model selection for time-series forecasting is a new setting, we adapt leading methods from algorithm selection, and include additional baselines by creating a variant of the proposed model (marked with TSL). Second, as shown in Table 14 we did a model selection on seven SOTA time series forecasting models as well (including DeepAR (2020), DeepFactors (ICML 2019), and Prophet (proposed by Facebook in 2018)) as baselines where we chose the model with the best average performance from all model variants. We updated Appendix I in which we explain the details of each baseline and discussion about our solution. Also, Appendix J.1 contains the aforementioned results of the seven forecasting models.
>
>
> Question 3: There are several meta-learning works for time-series forecasting missing, e.g. [R1, R2]. [R1] Pan, Zheyi, et al. "Spatio-temporal meta-learning for urban traffic prediction." IEEE Transactions on Knowledge and Data Engineering (2020). [R2] Narwariya, Jyoti, et al. "Meta-learning for few-shot time series classification." Proceedings of the 7th ACM IKDD CoDS and 25th COMAD. 2020. 28-36.
>
> Response: Thanks for pointing out those works, we did not come across these works while submitting the initial version. We have added citations to those works and mentioned the differences between our work and them in the revised version in the Related Work section. In particular, [R1] proposes meta-learning for the Spatio-Temporal traffic prediction problem and [R2] considers a few-shot time series classification problem and tackles only univariate datasets.
>
> Kindly let us know if our response addresses your concerns. We will be happy to answer if there are additional issues/questions.
>
>
> Sincerely,
>
> Authors of Paper 3114

---

### Decision · Program_Chairs · 2022-01-20

**Decision:**

Reject

**Comment:**

In this work the authors consider the automatic selection of time-series forecasting model (and hyperparameters) based on historical data. It adopts a conventional feature-based meta-learning approach. Experimental results show an improved performance over the considered baselines.

The reviewers appreciated the clarifications provided by the authors, but a number of concerns were unresolved. For instance, questions remained regarding the dataset collection, the baselines against which the proposed method was compared to (which were considered too weak) and the large number of missing details in the presentation of the method. Based on this the reviewers concluded that the paper could not be accepted in its current form and would require a major revision.